# Golden section criterion to achieve droplet trampoline effect on metal-based superhydrophobic surface

Shengteng Zhao [1], Zhichao Ma [1,2,3,4] ✉, Mingkai Song[1], Libo Tan[1], Hongwei Zhao [1,3,4] & Luquan Ren[2,4,5]

Clarifying the consecutive droplet rebound mechanisms can provide scientific inspirations to regulate dynamic wettability of superhydrophobic surface, which facilitates the practical applications on efficient heat control and active anti-icing. Generally, droplet rebound behaviors are directly affected by surface structure and Weber number. Here, we report a novel "golden section" design criterion to regulate the droplet rebound number determined by the structure spacing, subverting conventional knowledge. Especially, the droplet can continuously rebound for 17 times on the metal-based surface, exhibiting an amazing phenomenon of "droplet trampoline". The droplet rebound number has been experimentally revealed to be closely related to Weber number. We propose novel quantitative formulas to predict droplet rebound number and clarify the coupling effect of the structure spacing and the Weber number on the rebound mechanisms, which can be utilized to establish the regulation criteria of rebound numbers and develop novel metal-based superhydrophobic materials.

Since the equation of contact angle was proposed in 1805[1], the researches on surface wettability and solid-liquid interface behavior have been investigated for over 200 years, and are known as a "sleeping beauty" in science[2,3]. The discovery that organic compound adsorption layers can completely regulate the wettability of solid surfaces reveal the importance of chemical composition in determining the wettability[4]. The Wenzel model and the Cassie-Baxter model, which characterize the static contact angle of liquids on rough or structured surfaces, demonstrate the correlation between wettability and surface topography[5,6]. Numerous water-repellent surfaces have been prepared through constructing non-smooth structures and reducing the surface energy, benefiting from the development of micro-nano manufacturing technologies. Recently, the applications of superhydrophobic surfaces in the fields of droplet manipulation[7-9], droplet condensation[10-12], and anti-icing[13-15] have become research

hotspots. Meanwhile, the stability and durability of superhydrophobic surfaces[16-18], as well as the service performance of superhydrophobic surfaces under harsh conditions, are of great concern[19-21]. Especially for the scientific theories and experimental phenomena of surface wettability, researchers gradually focused on the dynamic wetting behavior of liquid droplets to match the dynamic control requirements of superhydrophobic surfaces in practical applications[22].

Many strategies have emerged to facilitate the rapid separation of droplets from superhydrophobic surfaces after revealing the interfering factors of droplet contact time[23-26]. For instance, the contact time can be reduced by 37% on a macroscale textured surface that divides the droplet into two parts[24]. The droplet exhibits an astonishing pancake bounce on the copper surface with tapered posts, and its contact time is ~80% shorter than the conventional contact time, breaking the Guinness record[25]. Inspired by Echeveria, the droplet

[1]School of Mechanical and Aerospace Engineering, Jilin University, Changchun 130025, China. [2]Key Laboratory of Bionic Engineering Ministry of Education, Jilin University, Changchun 130025, China. [3]Key Laboratory of CNC Equipment Reliability, Ministry of Education, Jilin University, Changchun 130025, China. [4]Institute of Structured and Architected Materials, Liaoning Academy of Materials, Shenyang 110167, China. [5]Weihai Institute for Bionics-Jilin University, Weihai 264207, China. ✉e-mail: zcma@jlu.edu.cn

contact time can be reduced by 40% by inducing asymmetric bounce of droplets on a corrugated surface[26]. Previous studies have mainly focused on the droplet contact time to evaluate the dynamic water repellency of surfaces. Nonetheless, the consecutive droplet rebound seems to be a potential phenomenon that has been ignored. In particular, the number of consecutive droplet rebounds ($N_R$) can substantially reflect the dynamic wettability of the surface, and exhibit a promisingly practical value in the fields of efficient heat control and active anti-icing. For example, droplet rebounds are expected to be suppressed on hot surfaces (inhibiting the Leidenfrost effect) to obtain high cooling efficiency[27], and are undesirable on plant surfaces for better pesticide absorption[28]. On the contrary, more droplet rebounds are required to remove more pollutants from surfaces to realize self-cleaning or to prevent droplets from staying on the surface to delay icing. Recently, multiple rebounds of droplet are reported on laser-ablated TC4 surfaces after organic adsorption and fluorination treatment[29], and the consecutive rebound characteristics of droplets on superhydrophobic surfaces are studied by analyzing the energy dissipation of droplets during the contact process[30]. However, the correlation between the consecutive rebound behavior of droplets and surface structure is crucial to regulate the surface dynamic wettability based on structure design. Moreover, the structure parameter and Weber number ($We$) are key factors that affect droplet dynamic behavior or surface wettability in most researches[8,12,15,19,23,25–27,29–31]. Hence, it is necessary to reveal the effect of surface structure spacing ($D_S$) on the $N_R$ on the basis of a wide range of $We$.

The design concept of fabricating fine microstructures and reducing the surface energy to obtain better hydrophobicity and facilitate droplets to detach from the surface is widely acknowledged. Conventionally, droplets can more easily rebound on the surface due to the smaller solid-liquid contact area of the denser microstructure corresponding to smaller $D_S$, as the microcolumn structure that induces pancake bouncing of droplets can significantly reduce the contact time by 80%[25]. However, the large dynamic pressure of small solid-liquid contact area may cause the permeation effect and the transition from Cassie-Baxter state to Wenzel state[32]. Here, we reveal that reducing the $D_S$ of microstructure is not the optimal solution to obtain higher $N_R$. The $D_S$ needs to be within an optimal medium range to obtain the maximum $N_R$, exhibiting a "golden section" effect of $N_R$. Especially, the maximum $N_R$ reached 17 times on aluminum-based surface in our experiments. The discovery is completely different from the design concept of superhydrophobic surface based on minimal microstructure, which subvert traditional cognition and can provide new horizons for superhydrophobic design. Meanwhile, the dynamic wettability based on $N_R$ highly depends on $We$, rather than the independence of conventional contact time with $We$[23]. Furthermore, different dynamics of droplets on microstructures with different $D_S$ leads to different $N_R$, relating to different energy dissipations during the dynamic bouncing process of droplets. We have deduced novel quantitative prediction formulas and distribution characteristics of $N_R$ based on $D_S$ and $We$ by introducing the variable $D_S$. The proposed theoretical model can be utilized to establish the regulation criteria of $N_R$ and is expected to inspire new strategies to realize efficient thermal management, anti-icing, self-cleaning and droplet control, etc.

## Results

### Different droplet dynamic behaviors

Generally, droplets will rebound after impacting, spreading and contracting on a superhydrophobic surface. A complete droplet rebound cycle includes four stages: falling, spreading, contracting and rising. Figure 1a–c show superhydrophobic laser-ablated surfaces composed of micro-protrusions and particles with $D_S$ of 50 μm, 500 μm and 1000 μm (referred to as S50, S500 and S1000 respectively). Water droplets (~4.5 μL) could repeatedly rebound on surfaces with these three typical $D_S$ in most cases. Figure 1d–f show that the droplets

continuously rebounded on three surfaces at a relatively lower $We$ of 22.2, with $N_R$ of 6, 17, and 6 respectively. It is difficult to confirm significant differences in the dynamic behavior of droplets on the three surfaces except for the values of $N_R$. However, the droplet rebound behaviors on the surfaces were different at a relatively higher $We$ of 61.0 (Fig. 1g–i). Specifically, the droplet was partially pinned to the microstructure and could not completely detach from the S50 surface with small $D_S$ (Fig. 1g), and there was liquid adhesion on the S1000 surface with large $D_S$ due to the large flat block (Fig. 1i). Briefly, both the partial pinning caused by small $D_S$ and the liquid adhesion caused by large $D_S$ prevented the droplet rebound, thereby limiting the $N_R$. Surprisingly, droplet could rebound for 12 times on S500 surface at high $We$ without any residual liquid, exhibiting excellent consecutive rebound ability. The droplet contact time was relatively stable as no pinning or adhesion occurred to delay the desorption between the droplet and surface, indicating that the S500 surface exhibited robust water repellency (Fig. 1e, h). Noteworthy is that more consecutive droplet rebounds appeared on surfaces with medium $D_S$ whether it was under low or high $We$. That means the $D_S$ can be optimized to obtain more consecutive droplet rebounds, which is expected to be explored.

### "Droplet trampoline" phenomenon

Especially, the droplet consecutively rebounded for 17 times on the S500 surface at $We = 22.22$ (Supplementary Fig. 1 and Supplementary Movie 2), demonstrating an amazing "droplet trampoline" phenomenon although metal-based surfaces are generally hydrophilic with surface energy difficult to be reduced. The droplet was first released and fell to contact the surface, during which time the gravitational potential energy was converted into kinetic energy. Then the droplet spread horizontally until it reached maximum transversal spreading due to the inertia force, and the kinetic energy was converted into surface energy. The droplet contracted under the action of surface tension, and the surface energy was converted to the kinetic energy, so the droplet obtained an upward speed and left the surface. And the kinetic energy was almost converted into gravitational potential energy when the droplet rose to the maximum height, completing one rebound cycle. During the contact process between the droplet and surface, there was energy dissipation accompanied by the conversion of kinetic energy and surface energy, so the kinetic energy of the droplet gradually attenuated until the kinetic energy was insufficient to support the droplet to separate from the surface. As the number of consecutive rebounds increased, the transversal spreading deformation of the droplet decreased due to the gradual decrease of the initial kinetic energy, resulting in a tendency of impact spheroidization.

### $N_R$ of microstructures with different $D_S$

In order to systematically explore the correlation between the $N_R$ and $D_S$, we prepared a series of surfaces with different $D_S$ by using laser ablation. We obtained a series of micro-protrusion structures with different $D_S$ by adjusting the laser scan spacing ranging from 50 μm to 1000 μm (referred to as S50-S1000). Micro-protrusions are arranged in square arrays with particles distributed on the laser ablation paths (Supplementary Figs 2–4), and the surface morphology with the scale of <1 μm is flocculent (Supplementary Fig. 5). A flat block appears on each micro-protrusion when the laser ablation area cannot completely cover the aluminum substrate (Supplementary Fig. 6). Furthermore, the larger laser scan spacing can raise the proportion of the flat area, and the flat block disappears if the laser scan spacing is small enough. In addition, the fluctuant heights corresponding to different laser scan spacings are similar (Supplementary Fig. 7). The laser scan speed which could change the structure height was constant in our experiment, and the change in laser scan spacing could not significantly affect the structure height. However, the structure height slightly changed which was inevitable due to the laser processing technology. But the

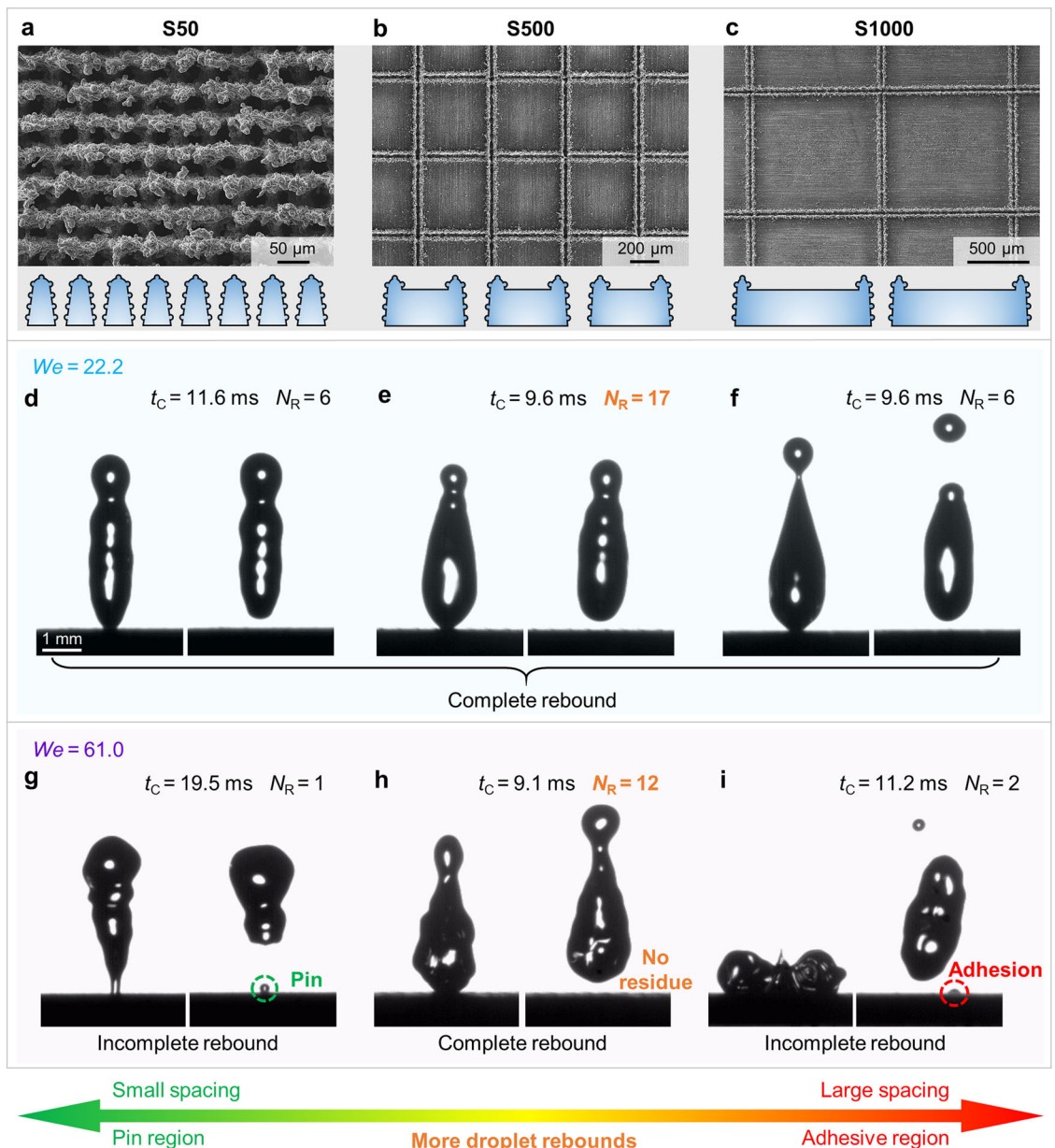

**Fig. 1 | Droplet dynamic behavior on laser-ablated microstructures with different $D_S$. a–c** Scanning electronic micrographs of laser-ablated aluminum surfaces with $D_S$ of 50 μm, 500 μm and 1000 μm, respectively. The micro-protrusions were arrayed in two perpendicular directions due to the grid paths of laser scanning, and particles induced by laser energy were deposited on the laser ablation lines. A flat block appeared on each micro-protrusion because the laser ablation area could not completely cover the aluminum substrate when the laser scan spacing was 500 μm or 1000 μm. **d–f** Similar droplet rebound behaviors on S50, S500 and S1000 surfaces at a relatively lower *We* of 22.2 (Supplementary Movies 1–3).

Droplets could completely rebound on all three surfaces without residue. The $N_R$ of S500 surface with medium $D_S$ was the highest although the $N_R$ of S50 and S1000 surfaces reached 6. **g–i** Different droplet rebound behaviors at a relatively higher *We* of 61.0 (Supplementary Movies 4-6). The droplet still completely rebounded on the S500 surface while liquid of the droplets remained on the S50 and S1000 surfaces. The $N_R$ of S500 was still high but the consecutive droplet rebound abilities of S50 and S1000 surfaces were greatly lost. Droplet contact times of S50 and S1000 increased obviously while the contact time of S500 was basically stable.

structure height hardly affected the consecutive rebound dynamics of droplets (Supplementary Information S1)[25,33–35]. The aluminum substrate was also thermally oxidized during the line-by-line laser scanning process, which could lead to the fact that oxidized aluminum films were formed. We discussed the localized X-ray diffraction results (Supplementary Fig. 8) on the laser-scanned grooves and flats in Supplementary Information S2.

We measured the $N_R$ of prepared microstructures with different $D_S$ in a wide range of *We* (Supplementary Fig. 9). As the $D_S$ continuously increased, the surfaces are divided into the pin region, gold region and adhesive region dependent on the characteristics of droplet rebound

behaviors. Specifically, the S300, S400 and S500 surfaces with no liquid residue are approximately at the golden ratio region in the overall $D_S$ range, which is named as the gold region. Figure 2a shows the concise $N_R$ characteristics of three regions. The surfaces of the gold region manifest as higher $N_R$ in a wide range of *We*, but the surfaces of the pin region and the adhesive region exhibit less $N_R$. Especially, the $N_R$ of the pin region obviously decreases, while the $N_R$ of the adhesive region fluctuate as the *We* continuously increases. We further extract the total average $N_R$ of each surface to compare the droplet rebound ability of surfaces with different $D_S$ (Fig. 2b). It is clearly visible that S300, S400 and S500 surfaces exhibit a superior high level of droplet

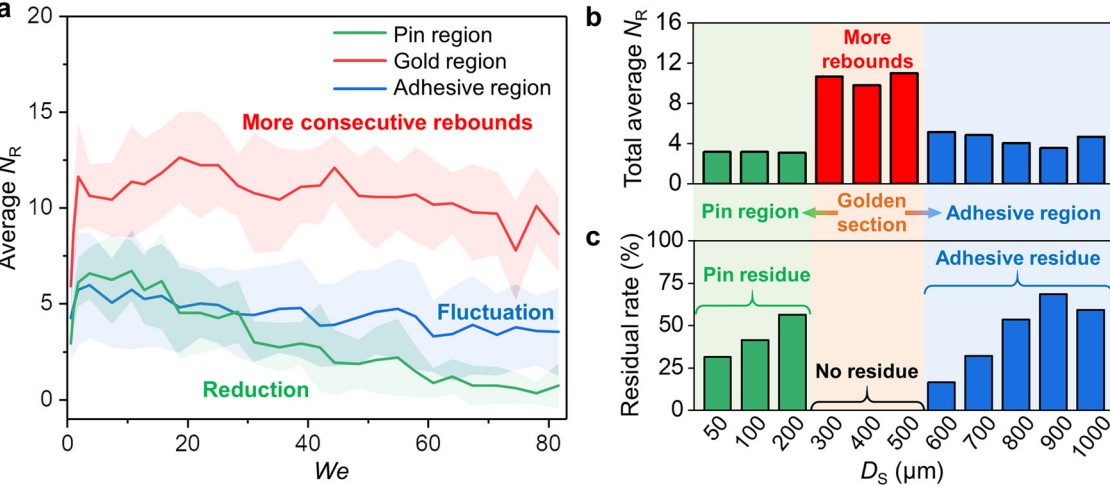

**Fig. 2 | Consecutive droplet rebound characteristics of laser-ablated structures with different $D_S$. a** Average $N_R$ of droplets impacting on surfaces of three typical regions on the basis of a series of $We$ in a range from 0 to 82. S300, S400 and S500 surfaces are in the gold region which manifests as obvious consecutive droplet rebounds. The pin region contains S50, S100 and S200 surfaces, and the adhesive region contains S600, S700, S800, S900 and S1000 surfaces. Each point denotes the average value of the surfaces of each region, and the error band of each point was obtained from the standard deviation of five replicate experiments for each surface (15 replicate experiments in total for pin region and gold region, and 25 replicate experiments in total for pin region). **b** The total average $N_R$ which represents the droplet rebound ability of each surface. Each value is calculated from points corresponding to all $We$ of one surface. **c** The residual rate of all surfaces. Laser-ablated microstructures with medium $D_S$ exhibited no residual liquid, and either too large or too small $D_S$ could facilitate the liquid residue. Especially, the surfaces in the gold region manifested as higher $N_R$ and no liquid residue, exhibiting a "golden section" effect.

rebound ability. In addition, we calculate the probability of incomplete rebound or failed rebound with residual liquid and obtain the residual rate of each surface (Fig. 2c). The residual rate decreases first and then increases with the gradual increase of $D_S$, which is opposite to the trend of the total average $N_R$. Therefore, the water repellency of the surface represented by consecutive droplet rebounds first enhances and then deteriorates as the $D_S$ increases. In particular, the residual rates of the surfaces of the gold region are all 0, exhibiting an amazing "golden section" effect.

### Difference in pin residue and adhesive residue

Pin residue of the droplet occurred on surfaces with small structure spacings due to the permeation effect on the microstructure caused by high dynamic pressure. The residual liquid with the contact angle >90° remained on the laser-ablated microstructure. Adhesive residue of the droplet occurred on surfaces with large structure spacings due to the adhesion effect of the large flat surrounded by microstructures. Large flats were induced by sparse laser ablation lines, which weakened the water-repellency of the surface. The residual liquid with the hydrophilic contact angle <90° remained on the flat that was not ablated. However, the medium structure spacing balanced the pinning effect and adhesion effect, avoiding liquid residue and greatly promoting the consecutive rebound of the droplet. Generally, reducing the solid-liquid contact area through microstructure is an indispensable condition for achieving superhydrophobicity. For droplets deposited statically ($We = 0$) on the surface or quasi-static impacted the surface at relatively low $We$, dense microstructures (small $D_S$) were beneficial for reducing solid-liquid contact area and achieving water-repellency. But at relatively high $We$, low solid-liquid contact area caused by dense microstructures could lead to significant wetting pressure. Dense undulations also increased the resistance of droplets to spread laterally, hindering the horizontal release of vertical impact force. Therefore, droplets were easier to penetrate and pin onto protruding particles in the impact area, and a portion of the liquid remained on the microstructure after rebound of the droplet. Increasing the spacing of microstructures, i.e. making them sparse, is an effective way to weaken the pinning effect. When the structure spacing increased to 300–500 μm, the consecutive rebound of the droplet was greatly

enhanced without any residue, as observed. However, as the structure spacing continued to increase, the laser-ablated microstructures that were beneficial for water-repellency became too sparse, resulting in large hydrophilic flats. The sparse microstructures and large flats seriously weakened the water-repellency of the surface and enhanced the affinity between the surface and water. The droplet could not completely detach from the surface due to the strong affinity of the large flat for water when rebounded from the surface. The liquid that could not be separated from the surface remained on the large hydrophilic flat. Hence, the small structure spacing and large structure spacing that caused the pin effect and adhesion effect to suppress the consecutive droplet rebound were the pin region and the adhesive region, respectively, and the medium structure spacing with high consecutive rebound times of the droplet and no liquid residue was the gold region.

## Discussion

### Analysis of droplet dynamics and $D_S$

In general, the dynamic behavior of droplets is directly related to the wettability of the surface. The surface hydrophobicity characterized by the contact angle and the sliding angle shows a monotonically decreasing trend with the continuous increase of $D_S$ (Supplementary Fig. 10)[31]. Evidentially, the surface wettability characterized by consecutive droplet rebounds is not monotonically changing as the $D_S$ increases, which is a new horizon in the design of superhydrophobic surfaces. Therefore, it is necessary to reveal the effect of $We$ and $D_S$ on $N_R$ through energy analysis. We clarify the main energy dissipation during the droplet dynamic process (Supplementary Information S2). The droplet gradually loses energy until the rebound kinetic energy is insufficient to support the droplet to separate from the surface. Thus, pin dissipation, viscous dissipation and adhesive dissipation significantly affect the $N_R$[36], and the energy dissipation characteristics of droplets determine the correlation between $N_R$ and $D_S$.

Firstly, with regard to the microstructure with small $D_S$, the small solid-liquid contact area increases the dynamic pressure of the droplet on the microstructure, and the particles on each micro-protrusion aggravate the contact pressure (Fig. 3a)[37]. Denser grooves increase the transversal spreading resistance of the droplet (Supplementary Fig. 11),

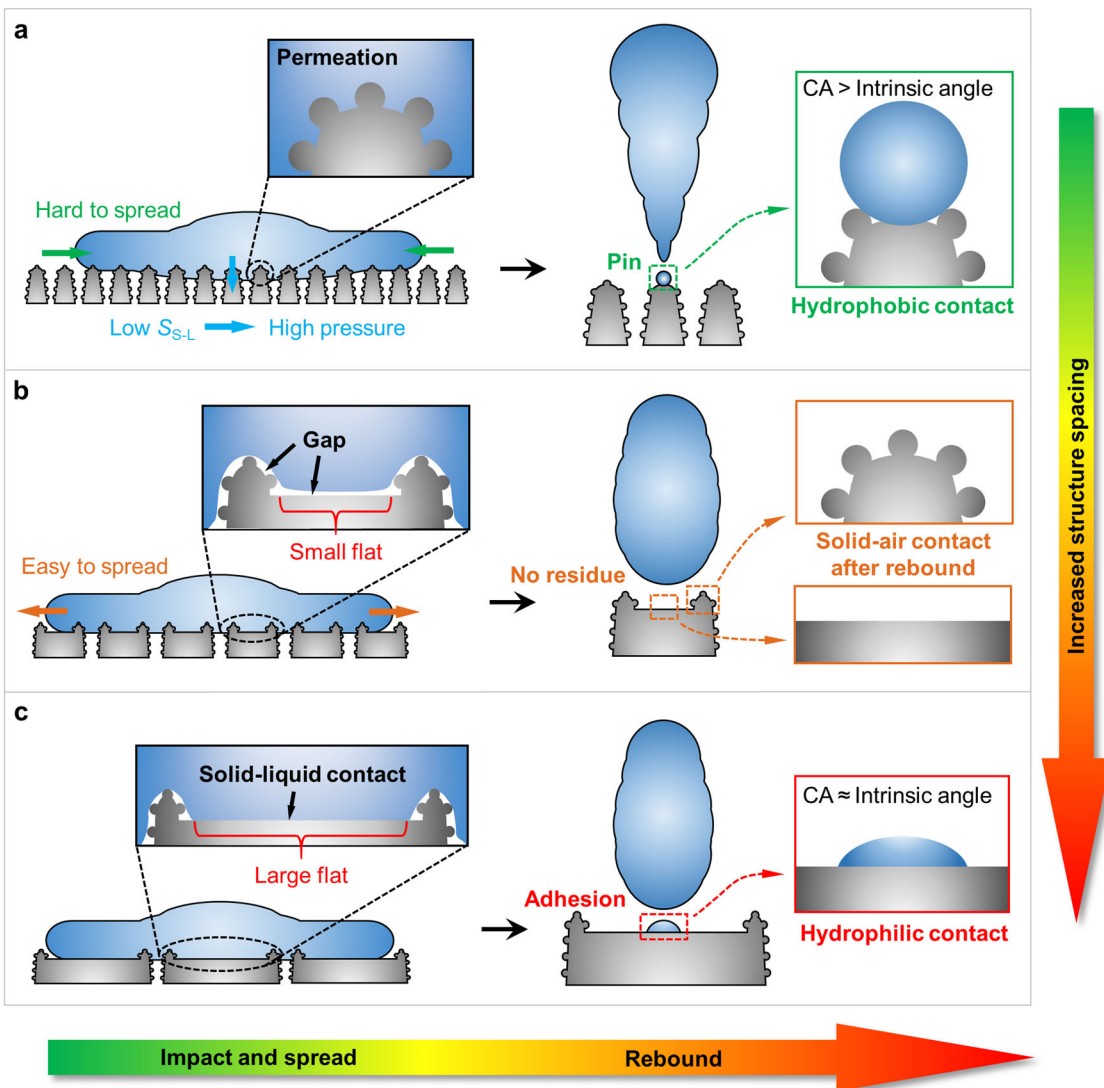

**Fig. 3 | Contact characteristics of solid-liquid interface and droplet dynamic behaviors on microstructures with different $D_S$. a** Permeation effect facilitated by high dynamic pressure due to large transversal resistance and small solid-liquid contact area ($S_{S-L}$) on the microstructure with small $D_S$. Trace amounts of liquid pin on the microstructure after the droplet separating from the surface. **b** Complete droplet rebound without residue on the microstructure with medium $D_S$. The vertical dynamic pressure can be released because the droplet is easy to spread horizontally and the $S_{S-L}$ is larger. The hierarchical superhydrophobic microstructure can suspend the liquid on the narrow flat although the flat is hydrophilic. **c** Solid-liquid interface adhesion on the hydrophilic large flat caused by large laser scan spacing. The surrounding superhydrophobic microstructure is not sufficient to support the liquid suspended on the wide hydrophilic flat, and the droplet is usually accompanied by residual liquid on the large flat when rebounding from the surface.

which hinders the transversal release of the solid-liquid dynamic pressure along the vertical direction. Droplets can easily penetrate into the grooves between particles under the promotion of high dynamic pressure due to high particle density, resulting in serious pin effect. Part of liquid mainly remains on protruding structures due to the pin effect after the droplet separating from the surface, causing incomplete droplet rebound. Because the droplet firstly contacted with the nearest protruding structure when it fell and reached the surface, and local solid-liquid contact on the protruding structure led to high wetting pressure and serious permeation effect. As the droplet laterally spread and infiltrated the grooves downwards, the solid-liquid contact area increased and the wetting pressure dispersed. Meanwhile, surface tension of the droplet prevented the liquid-air interface from penetrating deep grooves, which made it difficult for the droplet to contact grooves with high dynamic pressure and even diffuse into deep grooves. Meanwhile, the contact angle of the pinned droplet in experiments was usually much larger than the intrinsic contact angle

of aluminum, indicating that the penetration effect was the main reason for the droplet residue. Although the contact angles of the droplets with the Cassie-Baxter state and permeation state are both greater than the intrinsic contact angle of the surface. However, the droplet with the Cassie-Baxter state is easily separated from the surface while the pinned droplet with the permeation state is difficult to detach from the microstructure. Therefore, the contact angle of the droplet is greater than the intrinsic contact angle, but it cannot detach from the surface, indicating the occurrence of the partial penetration. Secondly, with regard to the microstructure with medium $D_S$, the reduced groove density can facilitate the spreading and contraction of droplets, which especially enhance the transversal release of vertical dynamic pressure of the droplet (Fig. 3b). Superhydrophobic microstructures around each micro-protrusion can support the liquid on the narrow hydrophilic flat, thereby avoiding the strong adhesion of the solid-liquid interface. The low energy loss and weak adhesion force facilitate the complete droplet rebound and induced higher $N_R$.

Thirdly, with regard to the microstructure with large $D_S$, the large hydrophilic flats greatly enhance the surface adhesion although the moving resistance of the solid-liquid contact line is lower (Fig. 3c). The span between microstructures around the flat is too long, which is insufficient to support the liquid and leads to the adhesive contact between the liquid and the flat. Therefore, some liquid remains on the hydrophilic flat due to the strong adhesion effect although the reverse kinetic energy can separate the liquid from the hydrophilic flat. Meanwhile, the contact angle of the residual liquid in experiments was close to the intrinsic contact angle of aluminum (Fig. 1i)[38], which was different from the permeation effect on the surface with small $D_S$, confirming that the strong hydrophilicity of the large flat caused the adhesion effect.

## Coupling effects of $We$ and $D_S$ on $N_R$

The restitution coefficient $\varepsilon = 1 - \frac{E_P + E_V + E_A}{E_K}$ is defined to represent the rebound kinetic energy of the droplet. We multiply the $\varepsilon$ by an amplification coefficient $N$ to evaluate the $N_R$. Therefore, the $N_R$ can be expressed as[30]:

$$N_R = N\varepsilon = N\left(1 - 0.284\frac{A_D}{D_S}We^{-0.5} - 0.055We^{0.75} - 1.14\times10^6 C_D D_F^2 We^{-1}\right)$$

$$(1)$$

where $A_D$ and $C_D$ denote correction factors of $D_S$ in pin dissipation and adhesive dissipation, respectively, and $D_F$ is defined as the width of the adhesive area on the hydrophilic flat. The $D_S$ of 500 μm was the critical structure spacing at which droplets could not statically slide on the surface, indicating that the hydrophilic flat was large enough to adsorb droplets and hinder their separation from the surface. Meanwhile, the adhesive area usually appeared only when the width of the hydrophilic flat increased to over the critical size (500 μm) according to the experimental results. Thus, $D_F$ can be expressed as $D_F = D_S\text{-}500$, and the $N_R$ can be further described as:

$$N_R = N\left(1 - \frac{A}{D_S}We^{-0.5} - BWe^{0.75} - C(D_S - 500)^2 We^{-1}\right) \quad (2)$$

where $A$, $B$, and $C$ are pre-factors of pin dissipation, viscous dissipation, and adhesive dissipation, respectively. The formula of $N_R$ only possesses two variables: $D_S$ and $We$. In particular, the expression is further expressed as $N_R = N(1 - \frac{A}{D_S}We^{-0.5} - BWe^{0.75})$ without considering adhesive dissipation when $D_S$ is less than the critical spacing.

The fitted pre-factor $A$ and $C$ are 50 and $1\times10^{-5}$, respectively (Supplementary Fig. 12). The fitted pre-factor $B$ of viscous dissipation for the surfaces of the pin region, gold region, and adhesive region are 0.03, 0.013 and 0.02, respectively, which are consistent with the calculated value of 0.055. The large friction and dynamic pressure of the structure with small $D_S$ aggravate the confusion of the internal flow field of the droplet, resulting in increased viscous dissipation, so the pre-factor $B$ is larger than the calculated value. The droplet manifests as less resistance when moving on the structure with medium $D_S$, and the internal flow field of the droplet is more orderly, so the viscous dissipation is weaker. The significant adhesion induced by large $D_S$ will aggravate the stickiness between the solid-liquid interface, causing a slight increase in the pre-factor $B$ of viscous dissipation. Figure 4a shows the dissipation characteristics of microstructures with different $D_S$. Especially, the pin dissipation is weakened and the adhesive dissipation is enhanced as $D_S$ increases. Therefore, pin dissipation and adhesive dissipation suppress the consecutive droplet rebound on structures with small $D_S$ and large $D_S$, respectively. We expressed the $N_R$ with the single variable $D_S$ to fit the total level of $N_R$, obtaining the prediction formula of the droplet rebound ability based on $D_S$ (Fig. 4b). The fitted line can provide a pathway to predict the droplet rebound ability on the basis of a given $D_S$. The theoretical model basically accords with the experimental results, even if the discreteness of $N_R$ and the significant randomness of fluid dynamics inevitably inhibit the consistence.

We obtain the coupled distribution of $N_R$ corresponding to different $We$ and $D_S$ by replacing the coefficient $N$ with the theoretical $N_R$ ability on the basis of different $D_S$. The distribution combines the $N_R$ level of different $D_S$ and the trend of $N_R$ along with $We$ (Fig. 4c). The theoretical distribution is substantially consistent with the experimental distribution of $N_R$ distribution (Fig. 4d), and contains a large amount of information. The highest $N_R$ corresponding to red color are concentrated in the low $We$ and medium $D_S$ region (left middle area), which suggested that multiple rebounds of droplet could be realized through designing a rational $D_S$ and a smaller $We$. We also mark out the region with significant large $N_R$ region in Supplementary Fig. 13, which is basically consistent with the theoretical distribution, indicating the feasibility of obtained distribution characteristics. However, large $We$ usually aggravates the energy loss of droplets or even causes the penetration into the surface, thus inhibiting the consecutive rebound of droplets (right area), which contradicts with the conventional concept that more impact energy of droplets is expected to obtain higher $N_R$[30]. Microstructures with small $D_S$ manifest as medium $N_R$ at low $We$ corresponding to stronger quasi-static superhydrophobicity (left bottom area). However, the penetration and pin effects induced by large solid-liquid dynamic pressure inhibit the consecutive rebound of droplets at high $We$ (right bottom area). With regard to the surfaces with large $D_S$, sparse laser-induced microstructures weaken the hydrophobicity. The droplets cannot even autonomously slide off the surface. Therefore, the droplet lacks enough kinetic energy to overcome the solid-liquid adhesion, inhibiting the rebound of the droplet at small $We$ corresponding to the weak hydrophobicity (top left area). The adhesive dissipation brought by the large $D_S$ limits the significant increase of $N_R$ although the consecutive rebound performance at high $We$ is better (top right area) with the enhanced droplet kinetic energy and small pin dissipation.

In summary, we report the novel "golden section" effect of the consecutive droplet rebound on laser-ablated microstructures with different $D_S$. The maximum $N_R$ reached 17, exhibiting the amazing "droplet trampoline" phenomenon on the aluminum-based surface. More consecutive droplet rebounds without residual liquid occurred on the surface with the $D_S$ in a range from 300 to 500 μm, inspiring the design criterion of a medium-spaced microstructure to enhance the droplet rebound ability. Pin dissipation and adhesive dissipation suppress the consecutive droplet rebound on microstructures with small $D_S$ and large $D_S$, respectively. We proposed a prediction formula and obtained the distribution map of $N_R$ on the basis of $We$ and $D_S$ with wide range. The derived theoretical model provides a golden section criterion to regulate the droplet rebounds, especially suitable for the realization of "droplet trampoline" effect on metal-based surface. For example, $We$ is usually difficult to be controlled on the metal-based surfaces with requirements of de-icing and self-cleaning, so the microstructure with medium $D_S$ should be designed to ensure the high droplet rebound ability available for a wide $We$ range. Conversely, most plant leaves are anti-wetting and their surface microstructures cannot be designed and fabricated. Pesticide utilization can be improved by regulating $We$ to inhibit droplet rebound, such as increasing the spraying speed (also improving the spraying efficiency) or modifying the pesticide liquid. For metal-based structures with medium spacing, consecutive rebounds of the droplet for more times can reduce the residence of the droplet on the surface, which can promote efficient transportation of droplets, and reduce the probability of surface icing and corrosion. Multiple contacts between the droplet and the surface can also remove more pollutants, thereby achieving better self-cleaning. The metal-based surfaces with more consecutive rebound times of the droplet have potential applications in the fields of aerospace, transportation and thermal management for

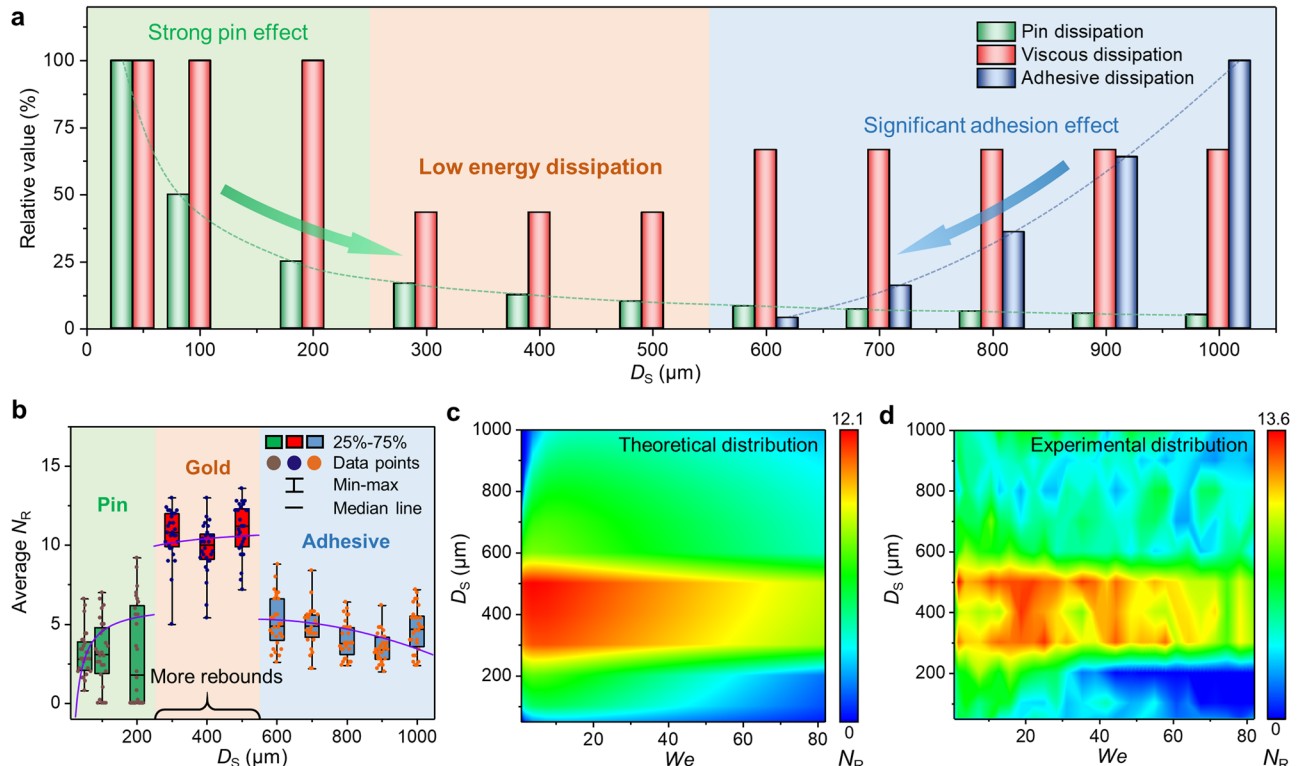

**Fig. 4 | The effect of $D_S$ on $N_R$ and the distribution map of $N_R$. a** Energy dissipation characteristics of all surfaces. The columns show the relative values of the three dissipations on the basis of different $D_S$, and the maximum value of each dissipation is regarded as 100%. **b** The analysis of correlation between $D_S$ and $N_R$. The equation of the fitted lines are $N_R = 9(1\text{-}20D_S^{-1}\text{-}0.3)$ for 50 μm ≤ $D_S$ < 300 μm, $N_R = 16(1\text{-}20D_S^{-1}\text{-}0.3)$ for 300 μm ≤ $D_S$ < 500 μm and $N_R = 8(1\text{-}20D_S^{-1}\text{-}0.3\text{-}1 \times 10^{-6}(D_S\text{-}500)^2)$ for $D_S$ ≥ 500 μm (also added adhesive dissipation for S600-S1000 surfaces). **c** Theoretical distribution of $N_R$ that combines the trend of $N_R$ with $We$ and the droplet rebound abilities of all surfaces on the coupled two dimensions of $We$ and

$D_S$ (several negative values are treated as 0). The trend of $N_R$ with $We$ is obtained by replacing the coefficient $N$ with the droplet rebound abilities corresponding to different $D_S$, and the droplet rebound abilities of microstructures with different $D_S$ are obtained from the fitted maximum value of average $N_R$. The equation of the fitted lines are $N_R = 15(1\text{-}20D_S^{-1}\text{-}0.3)$ for 50 μm ≤ $D_S$ < 300 μm, $N_R = 20(1\text{-}20D_S^{-1}\text{-}0.3)$ for 300 μm ≤ $D_S$ < 500 μm and $N_R = 13(1\text{-}20D_S^{-1}\text{-}0.3\text{-}1 \times 10^{-6}(D_S\text{-}500)^2)$ for $D_S$ ≥ 500 μm. **d** Experimental distribution of $N_R$ on the basis of two dimensions of $We$ and $D_S$.

anti-icing, anti-corrosion, self-cleaning and dropwise condensation. For metal-based structures with small and large spacing, the pin effect and adhesion effect can be used to capture trace volumes of droplets on the microstructure and flat, respectively, which have potential applications in fluid micromanipulation. Furthermore, droplet printing of selected areas can be achieved by combining structures with multiple spacings on one surface. Therefore, the proposed prediction method to regulate rebound numbers and behaviors of droplets can provide design criterion to develop novel metal-based superhydrophobic materials in various fields.

## Methods
### Materials and preparation
6061 aluminum sheets of 20 mm × 20 mm × 1 mm were selected as the substrates[39]. All aluminum sheets were mechanically polished and cleaned with absolute ethanol and deionized water before fabrication. Inspired by lotus leaves[40], nanosecond laser (F6020) with the wavelength of 1064 nm was used to irradiate aluminum substrates in grid paths. The diameter of the laser spot was 50 μm and the energy followed a Gaussian distribution. It is difficult to obtain typical microprotrusions with a laser scan spacing of <50 μm. Meanwhile, a spacing of 1000 μm corresponded to the radius of droplets with the volume of 4.2 μL, so the laser-ablated surface would lose anti-wetting property for most droplets with small volume if $D_S$ increased to over 1000 μm. Therefore, the scan spacings of grid paths were set as 50 μm, 100 μm, 200 μm, 300 μm, 400 μm, 500 μm, 600 μm, 700 μm, 800 μm, 900 μm and 1000 μm (referred to as S50, S100, S200, S300, S400, S500, S600,

S700, S800, S900 and S100, respectively), corresponding to different wettability of aluminum surfaces. The laser power, the scan speed and the repetition rate were set as 12 W, 100 mm/s and 20 kHz, respectively, to obtain a regular surface morphology with more particles[41]. Then the laser-ablated sheets were placed in an incubator at 260 °C to complete heat treatment for 6 h, which could reduce the free energy of aluminum surfaces[38]. Multiple samples were tested for each condition to ensure that the surfaces are not contaminated, damaged or otherwise altered in consecutive experiments. We also separately prepared different groups of specimens for multiple testing projects such as measurements of wettability, high-speed imaging of droplet dynamic behaviors, scanning electron microscopy imaging, laser confocal scanning of morphology and microfocus-beam X-ray diffraction to avoid mutual interference between different testing projects.

### Observation of microstructure
The morphologies of laser-ablated aluminum surfaces were observed by a field emission scanning electron microscope (Regulus 8100, Hitachi). Three-dimensional shapes and section profiles of the microprotrusions were observed by laser scanning confocal microscopy (OLS5100, Olympus). The geometric parameters of the microprotrusions were obtained by the average calculation method.

### Localized X-ray diffraction on the laser-scanned grooves and flats
Wide angle X-ray diffraction (WAXD) experiments were conducted on a customized microfocus X-ray diffraction system (Xenocs SA, France)

in transmission mode with Cu Kα radiation. The X-ray radiation wavelength was 0.154 nm, and the beam diameter of the X-ray radiation at the focal position was about 50 μm. Each WAXD pattern was exposed for 120 s at a sample-to-detector distance of 29 mm using a Pilatus 100 K detector (Dectris, Switzerland).

## Characterization of static wettability

The contact angle, slide angle and contact angle hysteresis were measured by a contact angle meter (JY-PHb, Chengde Jinhe Equipment Manufacturing Co, Ltd). The contact angle was measured with a 4 μL droplet of ultrapure water. The slide angle was the inclination angle of the surface when the 8 μL droplet of ultrapure water generated sliding displacement and finally left the surface. The advancing contact angle and receding contact angle were measured by increasing or decreasing the droplet volume.

## Characterization of droplet dynamics

The dynamics of the water droplets (with the diameter of $2.05 \pm 0.05$ mm) impacting on the superhydrophobic surfaces were captured by a high-speed camera (VEO-410L, Phantom) at a frame rate of 10000 fps[42]. The height of the needle from the surface was 3-130 mm considering the diameter of the droplet and the complexity of fluid flow at high impact velocity. The $We$ was calculated by the instantaneous velocity of the droplet when it contacted the surface. Five random measurements were carried out on each substrate.

## Count of droplet consecutive rebound numbers

The droplet observed to successfully separate from the surface by high-speed imaging was considered as one rebound, until the droplet separation from the surface could not be accurately identified. In the consecutive droplet rebound events, we divided two special cases of incomplete rebound: residual rebound and failed rebound. Specifically, residual rebound and failed rebound referred to the cases where a visible trace amount of liquid and a significant amount of liquid remained on the surface, respectively, after the first rebound of the droplet. The droplet rebounded for multiple times with the incomplete first rebound during residual rebound process, and the volume of the residual liquid was usually $2.7 \times 10^{-4}$-0.7 μL. During the failed rebound process, the droplet should adhere to the surface and cannot rebound during the failed rebound process, but the droplet might split into two secondary droplets containing the rebound droplet and the residual droplet during the rebound stage because the reverse kinetic energy was sufficient to overcome the surface tension of the droplet. The volume of the residual droplet adhering to the surface was usually >0.7 μL and the corresponding $N_R$ was 0. The ratio of residual rebound and failed rebound in all events of consecutive droplet rebound was defined as the residual rate.

## Energy analysis of droplet dynamics

The energy terms involved during the solid-liquid contact period of a droplet from the impacting to the detachment mainly include kinetic energy $E_K$, surface energy $E_S$, pin dissipation $E_P$, viscous dissipation $E_V$, and adhesive dissipation $E_A$ (only for surfaces of the adhesive region).

Firstly, the initial kinetic energy of a droplet can be expressed as $E_K = \frac{1}{12}\pi\rho D_0^3 V_1^2$, where $\rho$, $D_0$ and $V_1$ are the density, the diameter and the impact velocity of the droplet, respectively. The kinetic energy can be expressed by the variable $We = \frac{\rho D_0 V_1^2}{\gamma}$ as $E_K = \frac{1}{12}\pi D_0^2 \gamma We \sim We$. In our experiments, $D_0 = 2.05$ mm, and $\gamma = 72.75 \times 10^{-3}$ N/m. Therefore, the calculated prefactor is $8 \times 10^{-8}$, and $E_K \approx 8 \times 10^{-8} We$.

Secondly, the surface energy $E_S = \sum(\gamma S)$ is proportional to the surface area. As the surface area of the droplet increases due to the spreading, the kinetic energy is converted into the surface energy of the droplet. After the droplet contracts, the increased surface energy will be converted back into the rebound kinetic energy of the droplet as the surface area of the droplet decreases. We ignore the energy loss

of the conversion between kinetic energy and surface energy, and assume that the surface energy of the droplet before and after impacting the surface is equal. Therefore, the surface energy is only an intermediate product in our analysis.

Thirdly, laser ablation aggravates the surface roughness, resulting in the crawling movement of the contact line during the spreading and contracting process on the surface. Especially at the grooves induced by laser ablation, the surface has large fluctuations and obvious local wetting differences, causing the obvious pin effect of the contact line. Therefore, the droplet has the pin dissipation energy loss due to friction and critical resistance at grooves during the spreading and contracting. In order to characterize the critical resistance of the contact line at the laser-ablated groove, we measured the advancing contact angles ($\theta_A$) and receding contact angles ($\theta_R$) of microstructures with different $D_S$. As the $D_S$ increases continuously, the contact angle hysteresis shows an increasing trend, indicating the increased critical resistance at the laser-ablated groove. However, the pin dissipation of the droplet also depends on the density of the laser ablation grooves, that is also the $D_S$. Therefore, we add the $D_S$ and the correction factor $A_D$ to calculate the pin dissipation of the droplet during the spreading and contracting process:

$$E_P \approx \frac{A_D}{D_S}\left[\int_0^{R_{max}} 2\pi\gamma_{LA}(\cos\theta - \cos\theta_A)r\,\mathrm{d}r + \int_0^{R_{max}} 2\pi\gamma_{LA}(\cos\theta_R - \cos\theta)r\,\mathrm{d}r\right]$$
$$= \frac{A_D}{D_S}\pi\gamma_{LA}(\cos\theta_R - \cos\theta_A)R_{wetted}^2$$

$$(3)$$

where $\gamma_{LA} = 72.75 \times 10^{-3}$ N/m and $R_{wetted} \sim \frac{1}{2}D_0 We^{0.25}$ are the liquid-air interfacial tension and the maximum wetted radius of the droplet, respectively. The maximum wetted radius $R_{wetted}$ was close to $0.425 D_0 We^{0.25}$ in experiments as shown in Supplementary Fig. 11. For S50-S1000 surfaces, $\cos\theta_R$-$\cos\theta_A$ are 0.122, 0.110, 0.128, 0.143, 0.137, 0.145, 0.163, 0.207, 0.265, 0.392, 0.433, respectively, and the values of S50-S500 surfaces are very close. Large $D_S$ can weaken the difference of $\cos\theta_R$-$\cos\theta_A$, although the values of S600-S1000 surfaces in adhesive region significantly increase. Therefore, we substitute the average value 0.131 of S50-S500 surfaces into $\cos\theta_R - \cos\theta_A$ of all surfaces to simplify the calculation. The pin dissipation can be simplified as:

$$E_P \approx 2.27 \times 10^{-8}\frac{A_D}{D_S}We^{0.5}$$

$$(4)$$

Fourthly, viscous dissipation is generated due to the internal friction of the liquid, the edge vortex, plastic work, etc. in the process of droplet motion[36,43]. The viscous dissipation can be expressed as:[36]

$$E_V = \int_0^{t_C}\int_{\Omega_V}\mu\left(\frac{\partial\nu_i}{\partial x_j} + \frac{\partial\nu_j}{\partial x_i}\right)\frac{\partial\nu_i}{\partial x_j}\mathrm{d}t\mathrm{d}\Omega \approx \phi t_C\Omega_V$$

$$(5)$$

where $\phi \sim \mu\left(\frac{V_1}{L_1}\right)^2$, $t_C \sim \left(\frac{\rho D_0^3}{\gamma}\right)^{\frac{1}{2}} \approx 0.0109$ s and $\Omega_V = \pi L_V R_{wetted}^2$ are viscous dissipation function, the droplet contact time and the volume of viscous fluid, respectively. $\mu = 1 \times 10^{-3}$ Pa·s is the viscosity of the droplet. $L_V = 2\frac{D_0}{\sqrt{Re}}$ is the boundary layer thickness, and $Re = \frac{\rho D_0 V_1}{\mu} = We^{0.5}\frac{\sqrt{\rho D_0\gamma}}{\mu}$ is the Reynolds number. The viscous dissipation is calculated as:

$$E_V \approx 4.42 \times 10^{-9}We^{1.75}$$

$$(6)$$

Fifthly, adhesive dissipation (only for S600−S1000 surfaces) is the increase in interface energy of the system after adhesive rebound of

the droplet. The adhesive dissipation can be expressed as:

$$E_A = S(\gamma_{SA} + \gamma_{LA} - \gamma_{SL}) = S\gamma_{LA}(\cos\theta + 1) \tag{7}$$

where $S$, $\gamma_{SA}$, $\gamma_{SL}$ and $\theta$ are the adhesive area, the solid-air interfacial tension, the solid-liquid interfacial tension and the contact angle, respectively, and $\gamma_{SA} - \gamma_{SL} = \gamma_{LA}\cos\theta$. The final contact area is on a flat due to its hydrophilicity, when the droplet rebounds from the surface. Therefore, $\theta$ should be the contact angle ~75.3° of the smooth aluminum surface after heat treatment[38]. The adhesive dissipation is calculated as:

$$E_A \sim \gamma_{LA}(\cos\theta + 1)C_D D_F^2 \approx 9.12 \times 10^{-2} C_D D_F^2 \tag{8}$$

where $C_D$ is the correction factor, and $D_F$ is the width of the adhesive area on the hydrophilic flat.

In summary, we evaluate the $N_R$ by the restitution coefficient $\varepsilon$ based on kinetic energy $E_K$, pin dissipation $E_P$, viscous dissipation $E_V$, and adhesive dissipation $E_A$[30]:

$$\varepsilon = 1 - \frac{E_P + E_V + E_A}{E_K} \tag{9}$$

$$N_R = N\varepsilon = N\left(1 - 0.284\frac{A_D}{D_S}We^{-0.5} - 0.055We^{0.75} - 1.14 \times 10^6 C_D D_F^2 We^{-1}\right) \tag{10}$$

where $N$ is the amplification coefficient. The $N_R$ can be further simplified as:

$$N_R = N\left(1 - \frac{A}{D_S}We^{-0.5} - BWe^{0.75} - CD_F^2 We^{-1}\right) \tag{11}$$

$A$, $B$, and $C$ are integrated factors of the pin dissipation, viscous dissipation, and adhesive dissipation, respectively.

## Data availability

The main data supporting the findings of this study are available within the article and its Supplementary Information files. Source data are provided with this paper. All other relevant data supporting the findings of this study are available from the corresponding author on request.

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

## Acknowledgements

This work is funded by the National Natural Science Foundation of China (No. 52350039, No. 92266206 and No. 51925504), Jilin Province Science and Technology Development Plan (No. YDZJ202101ZYTS129), Jilin Province Creative and Innovative Talents Funding Project (No. 2023RY01), and Graduate Innovation Fund of Jilin University (No. 2022201). The authors are grateful for the support of laser ablation provided by Wenxuan Fan from Key Laboratory of Bionic Engineering Ministry of Education, Jilin University. The authors also gratefully thank Hongru Wang from Changchun Institute of Applied Chemistry, Chinese Academy of Sciences, and Wei Zhao and Guoxiang Shen from School of Mechanical and Aerospace Engineering, Jilin University for the helpful tests, analyses, and discussions of X-ray diffraction.

## Author contributions

S.Z. and Z.M. conceived the research and designed the experiments. S.Z. and M.S. carried out the experiments and developed the theoretical model. S.Z. and L.T. prepared the samples and analyzed the data. S.Z. wrote the manuscript with input from the other authors. Z.M. examined and polished the manuscript. Z.M., H.Z. and L.R. supervised the research. All authors contributed to the interpretation and drafting of the paper.

## Competing interests

The authors declare no competing interests.
