## [Peer Review File · Nature Communications]

REVIEWER COMMENTS

Reviewer #1 (Remarks to the Author):

The consecutive droplet rebound is taken as a new perspective on droplet impact in this work. The authors mainly study the droplet rebound number (NR) when the droplets impact on laser-ablated microstructures with different structure space (DS). In particular, NR even reach 17 on the S500 surface at $We=22.22$. To demonstrate the rebound mechanisms, the authors systematically explore the coupling effect of We and DS on NR, and then propose novel quantitative formulas to predict NR. This paper is interesting and with high quality. It can be published in Nature Communications after minor modification.

- 1) Although different laser scan spacings do not significantly change the structure height, the effect of the structure height on the rebound dynamics is unclear.
- 2) SEM images of the square area on the laser-ablated samples should be added, especially for medium and large DS. After all, the adhesion behavior is related to the square area (that is the large flat).
- 3) In 2.3, why does not the part of liquid remain on the grooves in Figure 3(a)?
- 4) For the wettability characterization in Figure S8, significant supporting images should be added.
- 5) The theoretical calculation of NR need to be further corrected since the experimental results is not well consistent with the theoretical model (Figure S11).
- 6) The authors are suggested to introduce a potential application of consecutive droplet rebound on the laser-ablated microstructures with certain range of DS.

This paper addresses an experimental characterization of consecutive droplet rebound on superhydrophobic aluminum-based surfaces, and qualitative/quantitative observation are made. Three different regions, namely, pin region, gold region and adhesive region can be compared to characterize the droplet rebound behaviors. Based on the energy analysis, the effect of We and Ds on the number of consecutive droplet rebound is discussed. And the authors developed a theoretical model to regulate the droplet rebounds.

The investigation appears to have some interests, and an amazing phenomenon of “droplet trampoline” that the droplet consecutively rebounds for 17 times is observed on the S500 surface at $We=22.22$. The authors stated that the occurrence of partial pinning or liquid adhesion prevented the droplet rebound, with well-explained quantitative data and a large amount of imagery from high-speed footage. They should take care to explain the theoretical reasoning for the occurrence of partial pinning or liquid adhesion.

However, there are some major outstanding issues with the description of results, strength of conclusion and method which would lead me to suggest that significant change are made to the paper before publication.

1. As the authors stated, the partial pinning caused by small Ds and the liquid adhesion caused by large Ds prevented the droplet rebound. However, the partially remained liquid is observed at both the pin region and the adhesive region. The authors should explain where they are making the difference. And the reasons for two region should be emphasized.
2. The author should give high-speed images to show the occurrence of partially remained liquid at the pin region due to the partial penetration. It is hard to follow the authors' descriptions of the high-speed footage.
3. The authors stated that the contact angle of the pinned droplet in experiments was usually much than the intrinsic contact angle of aluminum, indicating that the penetration effect was the main for the

droplet residue. However, if no partial penetration occurs, the contact angle of the pinned droplet is also than the intrinsic contact angle of aluminum, as the contact angle of the microstructure with small D_s is much than the intrinsic contact angle of aluminum.

4. As the authors stated, the amplification coefficient N is constant. However, the equations of the fitted line at the gold region match well the experimental results, and at the other region, the equation does not exhibit a good fit. It indicates whether the amplification coefficient N is not a constant.
5. A major concern is whether or not multiple samples were tested for each condition. As shown in the scanning electron microscope images of laser-ablated microstructures (Supplementary Fig. 2, 3 and 4), just one sample of each type was fabricated. For this reason, the authors need to reassure the readership that the surfaces are not contaminated, damaged or otherwise altered in consecutive experiments. Additionally, they should be wary of other factors such as wettability changes over time due to hydrocarbon adsorption from the laboratory's ambient air.
6. The microstructural characteristics is formed under the extreme condition where femtosecond laser shines on the aluminum-based surface with the ultrashort temporal scales ($\sim 10^{-15}$ s) and the high energy density ($> 10^{14}$ W/cm²). The flocculent structures are formed on the micro-protrusion due to the laser pulse ablation and the resolidification of ejected particles. The obvious differences between places near and far away the laser-scanned areas can be observed from the scanning electron microscope images of laser-ablated microstructures with large D_s (Supplementary Fig. 3 and 4). In addition, during the line-by-line laser scanning process, the aluminum substrate is also thermally oxidized, which will lead to the fact that oxidized aluminum films are formed. The authors should give the localized XRD results on the laser-scanned grooves and flat block.
7. The objective of section profiles in Supplementary Fig. 6c is to show that the structure height do not significantly change under different laser scan spacings. However, it is very difficult to get the data of

structure height from the figure by using normal linear ordinate. The authors should give a table to directly exhibit the geometry parameters.

8. The authors stated the “droplet trampoline” effect on metal-based superhydrophobic surface is investigated in this paper. However, there is no figure to show the superhydrophobicity of metal-based surfaces. In addition, is there a mode transition from Cassie-Baxter to Wenzel state with the continuous increase of D_s as the sudden increase of sliding angle occurs at the D_s of 500 μm .
9. Do the authors take into account the effect of energy dissipation caused by the partial penetration on the number of consecutive droplet rebounds. As the authors stated, the occurrence of partial penetration obviously inhibited the consecutive rebound of droplet.
10. The parameter of maximum wetted diameter (D_{wetted}) rather than the maximum spreading diameter (D_{max}) represent the diameter of pinned area, as the pin dissipation energy is originated from the crawling movement of the contact line during the spreading and contracting process.
11. As the authors stated, the parameter D_F in equation 6 (Supplementary Information S2) is the width of the adhesive area on the hydrophilic flat. How does the authors confirm that the critical size for the hydrophilic flat is 500 μm . In addition, the authors stated that the final contact area of residual liquid is hydrophilic region, which will raise a question whether the un-scanned areas remains intrinsic hydrophilicity.
12. When We exceeds 43, the number of consecutive droplet rebounds is zero on S200 surface (Supplementary Fig. 11), indicating that no successive droplet rebound occurs. The increase of We number obviously increases the dynamic pressure, which causes the fact that droplet can easily penetrate into the microstructures. The penetration effect produced by the increased We number is misleading qualitatively, which weakens their conclusion on S200 surface.

Point-by-point response to the reviewers' comments

Dear editor and reviewers,

We declare that the **submitted manuscript is the revised version of NCOMMS-23-25938-T**. Thank you very much for your review and valuable comments on our initial manuscript. We also appreciate the opportunities for the revision provided by the editor. Based on the comments, we made our best effort to revise our manuscript. We are so sorry to bring you so much trouble because of our carelessness. For your guidance, **itemized responses to all comments and corresponding revised contents** are appended below.

Reviewer #1 (Remarks to the Author):

The consecutive droplet rebound is taken as a new perspective on droplet impact in this work. The authors mainly study the droplet rebound number (NR) when the droplets impact on laser-ablated microstructures with different structure space (DS). In particular, NR even reach 17 on the S500 surface at $We=22.22$. To demonstrate the rebound mechanisms, the authors systematically explore the coupling effect of We and DS on NR, and then propose novel quantitative formulas to predict NR. This paper is interesting and with high quality. It can be published in Nature Communications after minor modification.

Response:

We thank the reviewer for the time spent on our paper and constructive comments. We are pleased that the reviewer found our work “interesting and with high quality”. Regarding the reviewer's concern about the experiments, discussions and the theoretical calculation of our article, we have substantially revised the manuscript and supplemented experiments to improve our study. We discussed in detail the effect of the structure height on the droplet rebound dynamics. Close-up scanning electron microscope images of the large flat on S1000 surface, which showed the morphology near and far from the laser-ablated grooves have been added. All electron microscopy images were re-captured using the field emission scanning electron microscope to improve resolution. Meanwhile, significant supporting images for the wettability characterization have been added. We have improved the discussion about droplet rebound dynamics on microstructures with different structure spacing and revised the theoretical calculation of the droplet rebound number. The theoretical model better matches the experimental results after revision. In addition, the potential applications of consecutive droplet rebound on the laser-ablated microstructures with certain range of structure spacing have been supplemented. The details of the revisions that we made are shown in the following responses, and changes to the text are highlighted in the manuscript.

Comment 1:

Although different laser scan spacings do not significantly change the structure height, the effect of the structure height on the rebound dynamics is unclear.

Response:

We appreciate the concern of the reviewer. The difference in structure height can affect the rebound

dynamics of droplets, especially their rebound behaviors. For example:

(1) Generally, an impinging droplet first spreads laterally to a maximum diameter, then retracts, and finally detaches from a superhydrophobic surface with relatively flat morphology, showing the conventional bouncing behavior. However, Liu et al found pancake bouncing as a novel droplet rebound behavior on copper substrate with micro-posts, which were fabricated by wire cut electrical discharge machining to achieve large aspect ratio. That is, when the structure height is large enough, the large aspect ratio structure can induce pancake bouncing of the droplet, significantly reducing the contact time and promoting the separation of the droplet from the surface. They also pointed out that when the structure height is sufficient to allow for adequate capillary energy storage, the emergence of pancake bouncing is rather insensitive to the post height. For much shorter posts, for example the tapered surface with a post height of 0.3 mm, they did not observe the pancake bouncing owing to insufficient energy storage. (Liu Y. H. et al. *Nat. Phys.* 10, 515-519, 2014).

(2) Pan et al prepared aluminum micro-conical pillar arrays with different heights by laser ablation, and studied the effect of the structure height on droplet rebound dynamics. They point out that impinging water droplets with different velocities generally show conventional bouncing processes with a constant contact time on a flat superhydrophobic surface. However, dynamic rebound processes of water droplets impinging on superhydrophobic micro-conical pillar arrays are different, and the contact time is significantly shortened. When the structure height was smaller than the critical value ($\sim 390 \mu\text{m}$), conventional bouncing processes with large contact time were observed. With the further increase of the structure height, pancake bouncing processes with the contact time smaller than 4 ms (approximately 27% of the conventional contact time) occurred on superhydrophobic micro-conical pillar arrays. Therefore, there exists a critical height of laser-ablated superhydrophobic aluminum structures, and a structure height higher than the critical height can induce the pancake bouncing of droplets. (Pan W. H. et al. *Nanoscale* 13, 14023-14034, 2021).

(3) In the research of Song et al, scan speed of the laser beam affects the structure height of laser-ablated aluminum surface, and the structure height decreases as the scan speed increases. They observed the rebound behavior of 2 μL droplets on the laser-ablated aluminum surface with scan speed of 50, 75, 100, 200 mm/s, respectively. It is shown that the rebound behavior is significantly influenced by the scan speed, which directly affects the structure height. When the scan speed is 50 mm/s, the water droplet can rebound from the surface twice due to the extremely low adhesion of the surface and the times of rebound decreases to one as the scan speed rise to 75 mm/s. Then, the droplet cannot leave the surface as the scan speed reaches to 100 mm/s or higher and it sticks to the surface, which indicates a high adhesion to droplet. These results further demonstrate that high structure can facilitate droplet rebound, and droplets cannot rebound on structures with too small height. (Song Y. X. et al. *Opt. Laser Technol.* 102, 25-31, 2018).

(4) Song et al systematically studied droplet rebound dynamics on shape memory polymer (SMP) pillars with commercial spray (Never Wet) as a non-metallic surface, including the effect of the structure height on droplet rebound behaviors. They showed the variation of the contact time of a water droplet on the superhydrophobic pillar arrays with different heights. The pancake bouncing was present at the structure height from 0.6 mm to 1.0 mm. However, for the structure height larger than 1.0 mm or smaller than 0.6 mm, the

droplet shows conventional bouncing. That means for non-metallic materials, structure height affects the droplet rebound behavior, and pancake bouncing appears only within a specific range of structure heights. (Song J. L. et al. ACS Nano 11, 9259-9267, 2017).

In summary, the structure height can qualitatively affect the droplet rebound behavior, and thus affect the contact time of the droplet. Specific structure height, especially large structure heights, can induce pancake bouncing with significant short contact time of the droplet, and the droplet cannot rebound on structures with too small height. The laser scan speed which could change the structure height was constant in our experiment, and the change in laser scan spacing could not significantly affect the structure height. However, structure height was slightly different in our experiments due to the laser processing technology. But our structure height almost all less than 100 μm , significantly lower than the structure height that induces pancake bouncing of droplet in above researches as shown in Table R1-1. Meanwhile, pancake bouncing did not occur on our laser-ablated structures, and the structures are high enough to support the rebound of the droplet. We focused on the effect of the structure spacing and We on the droplet rebound dynamics. Especially as extensive research has revealed the effect of the structure height (including metallic and non-metallic materials) on the droplet rebound dynamics, we did not repeat similar studies. In addition, we have supplemented structure height in the supplementary information for comparison.

The related researches are shown as follows:

Fig. R1.1. Influence of structure height of micro-conical pillar arrays on the contact time and bouncing shape of the droplet. (Pan W. H. et al. Nanoscale 13, 14023-14034, 2021)

Fig. R1.2. Left illustration (a): 3D profile of a typical surface, fabricated with the laser power of 8 W, a scan speed of 50 mm/s and scan interval of 50 μm . Right illustration (b): surface roughness of laser treated surfaces at different laser scan speeds. Bottom illustration (a)-(d): snapshots of a water droplet free-falling on the laser treated surfaces with different scan speeds of (a) 50 mm/s, (b) 75 mm/s, (c) 100 mm/s and (d) 200 mm/s. (Song Y. X. et al. Opt. Laser Technol. 102, 25-31, 2018)

Fig. R1.3. Variation of the contact time of a water droplet (17.9 μL) on the superhydrophobic pillar arrays with different height at $We = 13.3$ and the detachment moment of the water droplet on the superhydrophobic pillar arrays. (Song J. L. et al. ACS Nano 11, 9259-9267, 2017)

Table R1-1. Comparison of structure height between our research and others' research

Paper/Author	Structure height with pancake bouncing	Materials
Liu Y. H. et al. Nat. Phys. 10, 515-519, 2014	>300 μm	Copper
Pan W. H. et al. Nanoscale 13, 14023-14034, 2021	>390 μm	Aluminum
Song Y. X. et al. Opt. Laser Technol. 102, 25-31, 2018	No pancake bouncing Structure height: <100 μm	Aluminum
Song J. L. et al. ACS Nano 11, 9259-9267, 2017	600 μm -1000 μm	Shape memory polymer with Never Wet spray
This work	No pancake bouncing Structure height: <100 μm	Aluminum

The revised contents in section 2.2 of the main text are shown as:

In addition, the fluctuant heights corresponding to different laser scan spacings are similar (Supplementary Fig. 7). The laser scan speed which could change the structure height was constant in our experiment, and the change in laser scan spacing could not significantly affect the structure height. However, the structure height slightly changed which was inevitable due to the laser processing technology. But the structure height hardly affected the consecutive rebound dynamics of droplets (Supplementary Information S1). [25,33-35]

The optimized Supplementary Fig. 7 is shown as:

Supplementary Figure 7 | Geometric scanning of laser-ablated microstructures with a series of D_s . a-b, Three-dimensional image and section profile of S500 surface. The D_s was basically consistent with the set laser scanning spacing, and the average height difference between peak and valley of laser ablated grooves was 62.4 μm . c, Section profiles of microstructures with D_s of 50-1000 μm . Different laser scan spacings did not significantly change the structure height, and the slight differences in structure heights of different surfaces have almost no effect on the rebound dynamics of droplets.

The supplemented references are shown as:

- [25] Liu Y. H. et al. Pancake bouncing on superhydrophobic surfaces. *Nat. Phys.* 10, 515-519 (2014).
- [33] Pan, W. H., Wu, S., Huang, L., Song, J. L. Large-area fabrication of superhydrophobic micro-conical pillar arrays on various metallic substrates. *Nanoscale* 13, 14023-14034 (2021).
- [34] Song, Y. X. et al. Controllable superhydrophobic aluminum surfaces with tunable adhesion fabricated by femtosecond laser. *Opt. Laser Technol.* 102, 25-31 (2018).
- [35] Song, J. L. et al. Large-area fabrication of droplet pancake bouncing surface and control of bouncing state. *ACS Nano* 11, 9259-9267 (2017).

The supplemented contents in Supplementary Information S2 are shown as:

S2. The effect of the structure height on the rebound dynamics of droplets

Generally, an impinging droplet first spreads laterally to a maximum diameter, then retracts, and finally detaches from a superhydrophobic surface, showing the conventional bouncing behavior. However, pancake bouncing as a novel droplet rebound behavior has been found on copper substrate with micro-posts, which were fabricated by wire cut electrical discharge machining to achieve large aspect ratio. [1] The large aspect ratio structure can induce pancake bouncing of the droplet, significantly reducing the contact time and promoting the separation of the droplet from the surface when the structure height is large enough. Furthermore, when the structure height is sufficient to allow for adequate capillary energy storage, the emergence of pancake bouncing is rather insensitive to the post height. For much shorter posts, the pancake bouncing cannot be observed owing to insufficient energy storage. Similar results also appeared on aluminum micro-conical pillar arrays fabricated by laser ablation. [2] There exists a critical height of laser-ablated superhydrophobic aluminum structures, and a structure height higher than the critical height can induce the pancake bouncing of droplets. Conventional bouncing processes of the droplet with large contact time were observed when the structure height was smaller than the critical value. With the further increase of the structure height, pancake bouncing of the droplet with the contact time approximately 73% shorter than the conventional contact time occurred on superhydrophobic micro-conical pillar arrays.

High structure can induce the pancake bouncing of droplets, so how does the short structure affect the dynamics of droplet rebounds? Scan speed of the laser beam affects the structure height of laser-ablated aluminum surfaces, and the structure height can be reduced by increasing the scan speed. [3] The rebound behavior of droplet on short structures with the height less than 100 μm is significantly influenced by the scan speed, which directly affects the structure height. The water droplet can rebound from the surface twice due to the extremely low adhesion of the surface when the scan speed is low, and the times of rebound decreases to one as the scan speed rises. Then, the droplet cannot leave the surface as the scan speed continues to increase and it sticks to the surface, which indicates a high adhesion to droplet. Therefore, high structure can facilitate droplet rebound, and droplets cannot rebound on structures with too small height. Droplet rebound dynamics on shape memory polymer (SMP) pillars with commercial spray (Never Wet) as a non-metallic surface, including the effect of the structure height on droplet rebound behaviors have also been studied. [4] The pancake bouncing was present at the specific range of structure height. However, the droplet shows conventional bouncing when the structure height is larger or smaller than the critical values of the specific range. That means structure height affects the droplet rebound behavior, and pancake bouncing appears only within a specific range of structure heights.

In summary, the structure height can qualitatively affect the droplet rebound behavior, and thus affect the contact time of the droplet. Specific structure height, especially large structure heights, can induce pancake bouncing with significant short contact time of the droplet, and the droplet cannot rebound on structures with too small height. The laser scan speed which could change the structure height was constant in our experiment, and the change in laser scan spacing could not significantly affect the structure height. However, structure

height was slightly different in our experiments due to the laser processing technology. But the structure height almost all less than 100 μm , significantly lower than the structure height that induces pancake bouncing of droplet. Meanwhile, pancake bouncing did not occur on our laser-ablated structures, and the structures are high enough to support the rebound of the droplet.

Comment 2:

SEM images of the square area on the laser-ablated samples should be added, especially for medium and large DS. After all, the adhesion behavior is related to the square area (that is the large flat).

Response:

We thank the reviewer for the helpful suggestion. We have reshooted new SEM images with higher resolution of all surfaces with different spacings, including laser-ablated microstructures and flat areas that have not been ablated. Note that the diameter of the laser focal spot is 50 μm , so the laser ablated area almost completely cover the surface when the laser scan spacing is 50 μm . As shown in SEM images, there was no flat generated on S50 surface because all areas were ablated. Flats appeared due to the presence of areas that were not ablated when continue to increase the laser scan spacing. Meanwhile, the flat area became larger as the laser scan spacing increased, and the flats were significantly large when the laser scan spacing was set to 1000 μm . Scattered particles were sprayed onto the flats due to the jet effect of laser ablation. However, particles on flats were too sparse to significantly affect morphology and wettability of the surface. In addition, we have added close-up SEM images of the large flat on S1000 surface, showing the morphology near and far from the laser-ablated grooves.

New SEM images have been supplemented as shown in Supplementary Information:

Supplementary Figure 2 | Scanning electron microscope images of laser-ablated microstructures with D_s of 50 μm , 100 μm and 200 μm (pin region). Square-distributed arrays of micro-protrusions were formed

on the surfaces due to the grid shape of the laser scanning path. Particles were distributed along laser ablation paths and surrounded each micro-protrusion. There was no flat generated on S50 surface because all areas were ablated, and flats that were not ablated appeared on each micro-protrusion when the laser ablation lines were too sparse to completely cover the surface.

Supplementary Figure 3 | Scanning electron microscope images of laser-ablated microstructures with D_s of 300 μm , 400 μm and 500 μm (gold region). Small flats appeared on micro-protrusions because of the large laser scan spacing, and the flat area became larger as the laser scan spacing continuously increased.

Supplementary Figure 4 | Scanning electron microscope images of laser-ablated microstructures with D_s of 600 μm , 700 μm , 800 μm , 900 μm and 1000 μm (adhesive region). Large flats on micro-protrusions were significant because the laser scan spacing was too large, and the flats were extremely large when the laser scan spacing was set to 1000 μm .

Supplementary Figure 5 | High resolution scanning electron microscope images of laser-ablated microstructures with a series of D s. The flocculating morphology with scale less than 1 μm was formed due to the cooling of spray materials. No significant difference in morphology corresponding to different laser scan spacings.

Supplementary Figure 6 | Close-up scanning electron microscope images of the large flat on S1000 surface, showing the morphology near and far from the laser-ablated grooves. Scattered particles were

sprayed onto the flats due to the jet effect of laser ablation. However, particles on flats were too sparse to significantly affect morphology and wettability of the surface.

The revised contents in method of the main text are shown as:

The morphologies of laser-ablated aluminum surfaces were observed by a field emission scanning electron microscope (Regulus 8100, Hitachi).

Comment 3:

In 2.3, why does not the part of liquid remain on the grooves in Figure 3(a)?

Response:

We appreciate the reviewer's meticulousness. The droplet firstly contacted with the nearest protruding structure when it fell and reached the surface. Local solid-liquid contact on the protruding structure led to high wetting pressure and serious permeation effect. As the droplet laterally spread and infiltrated the grooves downwards, the solid-liquid contact area increased and the wetting pressure dispersed. Meanwhile, surface tension of the droplet prevented the liquid-air interface from penetrating deep grooves, which made it difficult for the droplet to contact grooves with high dynamic pressure and even diffuse into deep grooves. Therefore, serious permeation and droplet residue mainly occurred on protruding structures, especially the protruding structure within the area where droplet firstly contacted.

The revised contents in discussion of the main text are shown as:

Part of liquid mainly remains on protruding structures due to the pin effect after the droplet separating from the surface, causing incomplete droplet rebound. Because the droplet firstly contacted with the nearest protruding structure when it fell and reached the surface, and local solid-liquid contact on the protruding structure led to high wetting pressure and serious permeation effect. As the droplet laterally spread and infiltrated the grooves downwards, the solid-liquid contact area increased and the wetting pressure dispersed. Meanwhile, surface tension of the droplet prevented the liquid-air interface from penetrating deep grooves, which made it difficult for the droplet to contact grooves with high dynamic pressure and even diffuse into deep grooves.

Comment 4:

For the wettability characterization in Figure S8, significant supporting images should be added.

Response:

We thank the reviewer for the valuable comment. Significant supporting images of contact angle, slide angle, advancing contact angle and receding contact angle have been added to display wettability of surfaces with different spacings more intuitively.

The optimized Supplementary Fig. 10 is shown as:

Supplementary Figure 10 | Wettability characteristics of laser-ablated microstructures with different D_s . **a**, The reduced contact angle (CA) and the increased sliding angle (SA), indicating the weakened water repellency with the continuous increase of D_s . **b**, The advancing contact angle (ACA), receding contact angle (RCA) and calculated contact angle hysteresis (CAH). Large D_s aggravated the heterogeneity of wettability and increased the movement resistance of solid-liquid-air contact lines. **c-f**, Significant supporting images and results for the wettability characterization.

Comment 5:

The theoretical calculation of NR need to be further corrected since the experimental results is not well consistent with the theoretical model (Figure S11).

Response:

We thank the reviewer for the insightful comment. We have further corrected the theoretical calculation of N_R to make the theoretical lines better match the points according to the experimental results. Specifically, we corrected the coefficient N and the coefficient of viscous dissipation to better match the fitted lines with the experimental points in Supplementary Fig. 12 by mathematical fitting methods. The corrected theoretical calculation of N_R could well match the experimental results, demonstrating the availability of the formula. The amplification coefficient N was only a temporary constant in specific cases during single factor analysis, while N in the characterization formula of N_R for different We and D_S was a variable. The amplification coefficient N in the formula of N_R for different We and D_S was replaced by the average level of N_R corresponding to surfaces with different structure spacings. We validated the rationality of the proposed theoretical model based on We and D_S , respectively after deriving the physical formula of N_R . Firstly, we verified the trend of N_R increasing first and then decreasing as We continuously increased, as shown in the Supplementary Fig. 12. Meanwhile, the maximum average N_R of the S50-S500 surfaces increased sequentially, and the maximum average N_R of the S600-S1000 surfaces showed a decreasing trend as D_S continuously decreased. Secondly, we verified the trend of N_R increasing first and then decreasing as D_S continuously increased, as shown in the Fig. 4. In summary, the proposed theoretical model well described the effects of We and D_S on N_R .

In the revision, we made some corrections to the amplification coefficient N in the single factor analysis. Specifically, subtle modifications were made to the amplification coefficient N of each surface (D_S) to make the theoretical line more consistent with the experimental points based on each region having a specific N in the single factor analysis of We . Especially, the amplification coefficient N in the theoretical model of N_R was partitioned in the single factor analysis of D_S according to the comment. The amplification coefficient N was segmented in the pin area, the gold area and the adhesive area according to the experimental results, and the matching between the theoretical model and the experimental results was improved, as shown in Fig. 4. Inevitably, fluid experiments and the N_R in the experiments were all integers, resulting in high dispersion of the experimental results. The theoretical model based on physics was very difficult to fully match the experimental results. Nevertheless, we have revised the theoretical model and improved its compatibility with experimental results. The theoretical distribution of N_R better matched the experimental distribution of N_R corresponding to different We and D_S after revision.

The revised formulas are shown as:

$$N_R = N\varepsilon = N(1 - 0.284 \frac{A_D}{D_S} We^{-0.5} - 0.021We^{0.75} - 1.14 \times 10^6 C_D D_F^2 We^{-1}) \quad (1)$$

$$N_R = N(1 - \frac{A}{D_S} We^{-0.5} - BWe^{0.75} - C(D_S - 500)^2 We^{-1}) \quad (2)$$

The revised contents in discussion of the main text are shown as:

The fitted pre-factor A and C are 50 and 1×10^{-5} , respectively (Supplementary Fig. 12). The fitted pre-factor B of viscous dissipation for the surfaces of the pin region, gold region, and adhesive region are 0.03, 0.013 and 0.02, respectively, which are consistent with the calculated value of 0.021.

The revised Fig. 4 in the main text and Supplementary Fig. 12 are shown as:

Figure 4 | The effect of D_S on N_R and the distribution map of N_R . **a**, Energy dissipation characteristics of all surfaces. The columns show the relative values of the three dissipations on the basis of different D_S , and the maximum value of each dissipation is regarded as 100%. **b**, The analysis of correlation between D_S and N_R . The equation of the fitted lines are $N_R=9(1-20D_S^{-1}-0.3)$ for $50 \mu\text{m} \leq D_S < 300 \mu\text{m}$, $N_R=16(1-20D_S^{-1}-0.3)$ for $300 \mu\text{m} \leq D_S < 500 \mu\text{m}$ and $N_R=8(1-20D_S^{-1}-0.3-1 \times 10^{-6}(D_S-500)^2)$ for $D_S \geq 500 \mu\text{m}$ (also added adhesive dissipation for S600-S1000 surfaces). **c**, Theoretical distribution of N_R that combines the trend of N_R with We and the droplet rebound abilities of all surfaces on the coupled two dimensions of We and D_S (several negative values are treated as 0). The trend of N_R with We is obtained by replacing the coefficient N with the droplet rebound abilities corresponding to different D_S , and the droplet rebound abilities of microstructures with different D_S are obtained from the fitted maximum value of average N_R . The equation of the fitted lines are $N_R=15(1-20D_S^{-1}-0.3)$ for $50 \mu\text{m} \leq D_S < 300 \mu\text{m}$, $N_R=20(1-20D_S^{-1}-0.3)$ for $300 \mu\text{m} \leq D_S < 500 \mu\text{m}$ and $N_R=13(1-20D_S^{-1}-0.3-1 \times 10^{-6}(D_S-500)^2)$ for $D_S \geq 500 \mu\text{m}$. **d**, Experimental distribution of N_R on the basis of two dimensions of We and D_S .

Supplementary Figure 12 | The analysis of the correlation between We and N_R . The points originate from the experimental data, and the lines are fitted according to the formula of N_R . Each point is the average value of five points at the corresponding We in Supplementary Fig. 9. The fitted factor A is 50. The fitted factors B of viscous dissipation are 0.03, 0.013 and 0.02, respectively. The coefficients N of S50, S100 and S200 in pin region were 10, 9.5 and 10, respectively. The coefficients N of S300, S400 and S500 in gold region were 14.5, 14 and 14.5, respectively. The coefficients N of S600, S700, S800, S900 and S1000 in adhesive region were 8.5, 8, 7.8, 6 and 8.3, respectively. Especially, the adhesive dissipation only occurs on the surfaces of S600-S1000, and the factor $C=1 \times 10^{-5}$.

Comment 6:

The authors are suggested to introduce a potential application of consecutive droplet rebound on the laser-ablated microstructures with certain range of DS.

Response:

We agree with the reviewer's viewpoint. For structures with medium spacing, consecutive rebounds of the droplet for more times can reduce the residence of the droplet on the surface, which can promote efficient transportation of droplets, reduce the probability of surface icing and corrosion. Multiple contacts between the droplet and the surface can also remove more pollutants, thereby achieving better self-cleaning. The metal-based surfaces with more consecutive rebound times of the droplet have potential applications in the fields of aerospace, transportation and thermal management for anti-icing, anti-corrosion, self-cleaning and dropwise condensation. For structures with small and large spacing, the pin effect and adhesion effect can be used to capture trace volumes of droplets on the microstructure and flat, respectively, which have potential applications in fluid micromanipulation. Furthermore, droplet printing of selected areas can be achieved by combining structures with multiple spacings on one surface.

The revised contents in the end of the discussion (conclusion) of the main text are shown as:

For example, We is usually difficult to be controlled on the metal-based surfaces with requirements of de-icing and self-cleaning, so the microstructure with medium D_s should be designed to ensure the high droplet rebound ability available for a wide We range. Conversely, most plant leaves are anti-wetting and their surface microstructures cannot be designed and fabricated. Pesticide utilization can be improved by regulating We to inhibit droplet rebound, such as increasing the spraying speed (also improving the spraying efficiency) or

modifying the pesticide liquid. For metal-based structures with medium spacing, consecutive rebounds of the droplet for more times can reduce the residence of the droplet on the surface, which can promote efficient transportation of droplets, and reduce the probability of surface icing and corrosion. Multiple contacts between the droplet and the surface can also remove more pollutants, thereby achieving better self-cleaning. The metal-based surfaces with more consecutive rebound times of the droplet have potential applications in the fields of aerospace, transportation and thermal management for anti-icing, anti-corrosion, self-cleaning and dropwise condensation. For metal-based structures with small and large spacing, the pin effect and adhesion effect can be used to capture trace volumes of droplets on the microstructure and flat, respectively, which have potential applications in fluid micromanipulation. Furthermore, droplet printing of selected areas can be achieved by combining structures with multiple spacings on one surface. Therefore, the proposed prediction method to regulate rebound numbers and behaviors of droplets can provide design criterion to develop novel metal-based superhydrophobic materials in various fields.

Reviewer #2 (Remarks to the Author):

This paper addresses an experimental characterization of consecutive droplet rebound on superhydrophobic aluminum-based surfaces, and qualitative/quantitative observation are made. Three different regions, namely, pin region, gold region and adhesive region can be compared to characterize the droplet rebound behaviors. Based on the energy analysis, the effect of We and Ds on the number of consecutive droplet rebound is discussed. And the authors developed a theoretical model to regulate the droplet rebounds.

The investigation appears to have some interests, and an amazing phenomenon of “droplet trampoline” that the droplet consecutively rebounds for 17 times is observed on the S500 surface at $We=22.22$. The authors stated that the occurrence of partial pinning or liquid adhesion prevented the droplet rebound, with well-explained quantitative data and a large amount of imagery from high-speed footage. They should take care to explain the theoretical reasoning for the occurrence of partial pinning or liquid adhesion.

However, there are some major outstanding issues with the description of results, strength of conclusion and method which would lead me to suggest that significant change are made to the paper before publication.

Response:

We thank the reviewer for the time spent on our paper, constructive comments, and detailed remarks. We are very care for “major outstanding issues” with the description of results, strength of conclusion and method pointed out by the reviewer. Empowered by the reviewer's concern about the theoretical explanation, we have conducted extra experiments, polished our theoretical explanations and model, and carefully revised the manuscript according to the reviewer's comments. We believe that the scientific merit of our article is now much improved. We have supplemented a section in the main text to explain the difference between the pin residue and the adhesive residue at the pin region and the adhesive region, respectively. New high-speed movies of droplet dynamic behaviors on S50, S500 and S1000 surfaces at We of 22.2 and 61.0 have been added to enhance the persuasiveness of the article. Meanwhile, the structure heights of surfaces with different laser scan spacings have been added. We have made efforts to supplement the localized X-ray diffraction experiments on the laser-scanned grooves and flat block to reveal the thermal effect of laser ablation on aluminum materials. We also confirm that multiple samples were tested for each condition to ensure that the surfaces are not contaminated, damaged or otherwise altered in consecutive experiments. We have substantially revised the theoretical model of the droplet rebound number and replaced the maximum spreading diameter with the maximum wetted diameter. The theoretical model better matches the experimental results overall after revision. Besides, we have improved the discussions about droplet rebound dynamics on microstructures with different structure spacing, the penetration effect, the wettability and wetting state of laser-ablated surfaces and calculation formulas to strengthen readability and strictness of the article. The details of the revisions that we made are shown in the following responses, and changes to the text are highlighted in the manuscript.

Comment 1:

As the authors stated, the partial pinning caused by small Ds and the liquid adhesion caused by large Ds

prevented the droplet rebound. However, the partially remained liquid is observed at both the pin region and the adhesive region. The authors should explain where they are making the difference. And the reasons for two region should be emphasized.

Response:

We thank the reviewer for the valuable comment. Pin residue of the droplet occurred on surfaces with small structure spacings due to the permeation effect on the microstructure caused by high dynamic pressure. The residual liquid with the contact angle $>90^\circ$ remained on the laser-ablated microstructure. Adhesive residue of the droplet occurred on surfaces with large structure spacings due to the adhesion effect of the large flat surrounded by microstructures. Large flats were induced by sparse laser ablation lines, which weakened the water-repellency of the surface. The residual liquid with the hydrophilic contact angle $<90^\circ$ remained on the flat that was not ablated. However, the medium structure spacing balanced the pinning effect and adhesion effect, avoiding liquid residue and greatly promoting the consecutive rebound of the droplet.

Generally, reducing the solid-liquid contact area through microstructure is an indispensable condition for achieving superhydrophobicity. For droplets deposited statically ($We=0$) on the surface or quasi-static impacted the surface at relatively low We , dense microstructures (small D_s) were beneficial for reducing solid-liquid contact area and achieving water-repellency. But at relatively high We , low solid-liquid contact area caused by dense microstructures could lead to significant wetting pressure. Dense undulations also increased the resistance of droplets to spread laterally, hindering the horizontal release of vertical impact force. Therefore, droplets were easy to penetrate and pin onto protruding particles in the impact point area, and a portion of the liquid remained on the microstructure after rebound of the droplet.

Increasing the spacing of microstructures, i.e. making them sparse, is an effective way to weaken the pinning effect. When the structure spacing increased to 300-500 μm , the consecutive rebound of the droplet was greatly enhanced without any residue, as observed. However, as the structure spacing continued to increase, the laser-ablated microstructures that were beneficial for water-repellency became too sparse, resulting in large hydrophilic flats. The sparse microstructures and large flats seriously weakened the water-repellency of the surface and enhanced the affinity between the surface and water. The droplet could not completely detach from the surface due to the strong affinity of the large flat for water when rebounded from the surface. The liquid that could not be separated from the surface remained on the large hydrophilic flat.

Hence, the small structure spacing and large structure spacing that caused the pin effect and adhesion effect to suppress the consecutive droplet rebound were the pin region and the adhesive region, respectively, and the medium structure spacing with high consecutive rebound times of the droplet and no liquid residue was the gold region.

The revised contents in the results of the main text are shown as:

Pin residue of the droplet occurred on surfaces with small structure spacings due to the permeation effect on the microstructure caused by high dynamic pressure. The residual liquid with the contact angle $>90^\circ$

remained on the laser-ablated microstructure. Adhesive residue of the droplet occurred on surfaces with large structure spacings due to the adhesion effect of the large flat surrounded by microstructures. Large flats were induced by sparse laser ablation lines, which weakened the water-repellency of the surface. The residual liquid with the hydrophilic contact angle $<90^\circ$ remained on the flat that was not ablated. However, the medium structure spacing balanced the pinning effect and adhesion effect, avoiding liquid residue and greatly promoting the consecutive rebound of the droplet. Generally, reducing the solid-liquid contact area through microstructure is an indispensable condition for achieving superhydrophobicity. For droplets deposited statically ($We=0$) on the surface or quasi-static impacted the surface at relatively low We , dense microstructures (small D_s) were beneficial for reducing solid-liquid contact area and achieving water-repellency. But at relatively high We , low solid-liquid contact area caused by dense microstructures could lead to significant wetting pressure. Dense undulations also increased the resistance of droplets to spread laterally, hindering the horizontal release of vertical impact force. Therefore, droplets were easier to penetrate and pin onto protruding particles in the impact area, and a portion of the liquid remained on the microstructure after rebound of the droplet. Increasing the spacing of microstructures, i.e. making them sparse, is an effective way to weaken the pinning effect. When the structure spacing increased to 300-500 μm , the consecutive rebound of the droplet was greatly enhanced without any residue, as observed. However, as the structure spacing continued to increase, the laser-ablated microstructures that were beneficial for water-repellency became too sparse, resulting in large hydrophilic flats. The sparse microstructures and large flats seriously weakened the water-repellency of the surface and enhanced the affinity between the surface and water. The droplet could not completely detach from the surface due to the strong affinity of the large flat for water when rebounded from the surface. The liquid that could not be separated from the surface remained on the large hydrophilic flat. Hence, the small structure spacing and large structure spacing that caused the pin effect and adhesion effect to suppress the consecutive droplet rebound were the pin region and the adhesive region, respectively, and the medium structure spacing with high consecutive rebound times of the droplet and no liquid residue was the gold region.

Comment 2:

The author should give high-speed images to show the occurrence of partially remained liquid at the pin region due to the partial penetration. It is hard to follow the authors' descriptions of the high-speed footage.

Response:

We appreciate the concern of the reviewer. We have supplemented high-speed movies of droplet dynamic behaviors on S50, S500 and S1000 surfaces at We of 22.2 and 61.0 corresponding to Fig. 1. Afterwards, the process of liquid residue induced by pin and adhesion can be observed and compared more clearly. **Five new movies have been added** and the number of supplementary movies has increased to six based on the guidance of the reviewer, and **the six supplementary movies were cited in the caption of Fig. 1.**

Comment 3:

The authors stated that the contact angle of the pinned droplet in experiments was usually much smaller than the intrinsic contact angle of aluminum, indicating that the penetration effect was the main for the droplet residue.

However, if no partial penetration occurs, the contact angle of the pinned droplet is also than the intrinsic contact angle of aluminum, as the contact angle of the microstructure with small D_s is much than the intrinsic contact angle of aluminum.

Response:

We agree with the reviewer and thank the important comment of the reviewer. As the reviewer stated, the contact angles of droplets on the microstructure, whether pinned in the Wenzel state or deposited in the Cassie-Baxter state, are all greater than the intrinsic contact angle of the aluminum surface. Simply stating that the contact angle of the pinned droplet is greater than the intrinsic contact angle of aluminum is not sufficient to indicate the occurrence of penetration.

The penetration induced by wetting pressure can cause the transition of the contact state between droplets and local particles from the Cassie-Baxter state to the permeation state. Generally, the contact angle gradually decreases, but still higher than the intrinsic contact angle of the surface during the transition of droplet contact state from Cassie-Baxter state to the permeation state on a superhydrophobic surface (Lafuma, A., Quere, D. *Nat. Mater.* 2, 457-460, 2003).

The contact angles of the droplets with the Cassie-Baxter state and permeation state are both greater than the intrinsic contact angle of the surface. However, the droplet with the Cassie-Baxter state is easily separated from the surface while the pinned droplet with the permeation state is difficult to detach from the microstructure. Therefore, the contact angle of the droplet is greater than the intrinsic contact angle, but it cannot detach from the surface, indicating the occurrence of the partial penetration. In our experiment, another residue induced by adhesion effect occurred on the surface with large structure spacing. For the liquid residue induced by adhesion effect on the surface with large structure spacing, the contact angle of the adhered droplet was close to the intrinsic contact angle of the surface, which was different from the permeation effect on the surface with small structure spacing.

The previous description of the penetration effect is confusing to readers, and we have improved the confusing contents. **The revised contents in discussion of the main text are shown as:**

Meanwhile, the contact angle of the pinned droplet in experiments was usually much larger than the intrinsic contact angle of aluminum, indicating that the penetration effect was the main reason for the droplet residue. Although the contact angles of the droplets with the Cassie-Baxter state and permeation state are both greater than the intrinsic contact angle of the surface. However, the droplet with the Cassie-Baxter state is easily separated from the surface while the pinned droplet with the permeation state is difficult to detach from the microstructure. Therefore, the contact angle of the droplet is greater than the intrinsic contact angle, but it cannot detach from the surface, indicating the occurrence of the partial penetration.

Meanwhile, the contact angle of the residual liquid in experiments was close to the intrinsic contact angle of aluminum (Fig. 1i), [32] which was different from the permeation effect on the surface with small D_s , confirming that the strong hydrophilicity of the large flat caused the adhesion effect.

Comment 4:

As the authors stated, the amplification coefficient N is constant. However, the equations of the fitted line at the gold region match well the experimental results, and at the other region, the equation does not exhibit a good fit. It indicates whether the amplification coefficient N is not a constant.

Response:

We thank the reviewer for the insightful comment. The amplification coefficient N was only a temporary constant in specific cases during single factor analysis, while N in the characterization formula of N_R for different We and D_S was a variable. The amplification coefficient N in the formula of N_R for different We and D_S was replaced by the average level of N_R corresponding to surfaces with different structure spacings. We validated the rationality of the proposed theoretical model based on We and D_S , respectively after deriving the physical formula of N_R . Firstly, we verified the trend of N_R increasing first and then decreasing as We continuously increased, as shown in the Supplementary Fig. 12. Meanwhile, the maximum average N_R of the S50-S500 surfaces increased sequentially, and the maximum average N_R of the S600-S1000 surfaces showed a decreasing trend as D_S continuously decreased. Secondly, we verified the trend of N_R increasing first and then decreasing as D_S continuously increased, as shown in the Fig. 4. In summary, the proposed theoretical model well described the effects of We and D_S on N_R .

In the revision, we made some corrections to the amplification coefficient N in the single factor analysis. Specifically, subtle modifications were made to the amplification coefficient N of each surface (D_S) to make the theoretical line more consistent with the experimental points based on each region having a specific N in the single factor analysis of We . Especially, the amplification coefficient N in the theoretical model of N_R was partitioned in the single factor analysis of D_S according to the comment. The amplification coefficient N was segmented in the pin area, the gold area and the adhesive area according to the experimental results, and the matching between the theoretical model and the experimental results was improved, as shown in Fig. 4. Inevitably, fluid experiments and the N_R in the experiments were all integers, resulting in high dispersion of the experimental results. The theoretical model based on physics was very difficult to fully match the experimental results. Nevertheless, we have revised the theoretical model and improved its compatibility with experimental results. The theoretical distribution of N_R better matched the experimental distribution of N_R corresponding to different We and D_S after revision.

The revised formulas are shown as:

$$N_R = N\varepsilon = N(1 - 0.284\frac{A_D}{D_S}We^{-0.5} - 0.021We^{0.75} - 1.14 \times 10^6 C_D D_F^2 We^{-1}) \quad (1)$$

$$N_R = N(1 - \frac{A}{D_S}We^{-0.5} - BWe^{0.75} - C(D_S - 500)^2 We^{-1}) \quad (2)$$

The revised contents in discussion of the main text are shown as:

The fitted pre-factor A and C are 50 and 1×10^{-5} , respectively (Supplementary Fig. 12). The fitted pre-factor B of viscous dissipation for the surfaces of the pin region, gold region, and adhesive region are 0.03, 0.013 and

0.02, respectively, which are consistent with the calculated value of 0.021.

The revised Fig. 4 in the main text and Supplementary Fig. 12 are shown as:

Figure 4 | The effect of D_s on N_R and the distribution map of N_R . **a**, Energy dissipation characteristics of all surfaces. The columns show the relative values of the three dissipations on the basis of different D_s , and the maximum value of each dissipation is regarded as 100%. **b**, The analysis of correlation between D_s and N_R . The equation of the fitted lines are $N_R=9(1-20D_s^{-1}-0.3)$ for $50 \mu\text{m} \leq D_s < 300 \mu\text{m}$, $N_R=16(1-20D_s^{-1}-0.3)$ for $300 \mu\text{m} \leq D_s < 500 \mu\text{m}$ and $N_R=8(1-20D_s^{-1}-0.3-1 \times 10^{-6}(D_s-500)^2)$ for $D_s \geq 500 \mu\text{m}$ (also added adhesive dissipation for S600-S1000 surfaces). **c**, Theoretical distribution of N_R that combines the trend of N_R with We and the droplet rebound abilities of all surfaces on the coupled two dimensions of We and D_s (several negative values are treated as 0). The trend of N_R with We is obtained by replacing the coefficient N with the droplet rebound abilities corresponding to different D_s , and the droplet rebound abilities of microstructures with different D_s are obtained from the fitted maximum value of average N_R . The equation of the fitted lines are $N_R=15(1-20D_s^{-1}-0.3)$ for $50 \mu\text{m} \leq D_s < 300 \mu\text{m}$, $N_R=20(1-20D_s^{-1}-0.3)$ for $300 \mu\text{m} \leq D_s < 500 \mu\text{m}$ and $N_R=13(1-20D_s^{-1}-0.3-1 \times 10^{-6}(D_s-500)^2)$ for $D_s \geq 500 \mu\text{m}$. **d**, Experimental distribution of N_R on the basis of two dimensions of We and D_s .

Supplementary Figure 12 | The analysis of the correlation between We and N_R . The points originate from the experimental data, and the lines are fitted according to the formula of N_R . Each point is the average value of five points at the corresponding We in Supplementary Fig. 9. The fitted factor A is 50. The fitted factors B of viscous dissipation are 0.03, 0.013 and 0.02, respectively. The coefficients N of S50, S100 and S200 in pin region were 10, 9.5 and 10, respectively. The coefficients N of S300, S400 and S500 in gold region were 14.5, 14 and 14.5, respectively. The coefficients N of S600, S700, S800, S900 and S1000 in adhesive region were 8.5, 8, 7.8, 6 and 8.3, respectively. Especially, the adhesive dissipation only occurs on the surfaces of S600-S1000, and the factor $C=1 \times 10^{-5}$.

Comment 5:

A major concern is whether or not multiple samples were tested for each condition. As shown in the scanning electron microscope images of laser-ablated microstructures (Supplementary Fig. 2, 3 and 4), just one sample of each type was fabricated. For this reason, the authors need to reassure the readership that the surfaces are not contaminated, damaged or otherwise altered in consecutive experiments. Additionally, they should be wary of other factors such as wettability changes over time due to hydrocarbon adsorption from the laboratory’s ambient air.

Response:

We appreciate the necessary concerns of the reviewer. In our experiments, we separately prepared different groups of specimens for multiple testing projects such as measurements of wettability, high-speed imaging of droplet dynamic behaviors, scanning electron microscopy imaging, laser confocal scanning of morphology and microfocus-beam X-ray diffraction. This can avoid mutual interference between different testing projects. For example, electron beams during scanning electron microscopy imaging may affect the crystal characteristics or chemical composition of the substrate, thereby affecting X-ray diffraction testing and wettability of the sample, respectively.

The samples may be contaminated, damaged or otherwise altered in consecutive experiments as stated by the reviewer, which is very reasonable. Especially, the hydrophobicity of the surface can be weakened after pin or adhesion of droplets at relatively high We . Therefore, multiple samples were tested for each condition in characterizations of wettability and droplet dynamics, which has been sufficiently considered in previous experiments. Meanwhile, we have changed the droplet impact point for each high-speed imaging to suppress

the contingency of experimental results in previous experiments.

The wettability of laser-ablated aluminum microstructure can transform from hydrophilic to hydrophobic due to hydrocarbon adsorption from the ambient air. However, the superhydrophobic wettability of the laser-ablated aluminum surface is difficult to change when the adsorption is almost saturated. Normally, an aluminum surface after laser ablation consists of laser-ablated microstructures and flat areas that have not been ablated. On the one hand, all laser-ablated aluminum surfaces had completed sufficient organic adsorption process after heat treatment, and hydrocarbon adsorption is difficult to be sustained to change the superhydrophobicity of laser-ablated microstructures, which has been confirmed in previous studies (Zhao S. T. et al. Mater. Des. 223, 111145, 2022) (Zhao, S. T. et al. Appl. Surf. Sci. 611, 155652, 2023). On the other hand, organic pollution in the air cannot significantly affect the wettability of flat aluminum surfaces, and the hydrophilicity and adhesion of flat area is difficult to change even when exposed to air, which has also been confirmed (Zhao S. T. et al. Mater. Des. 223, 111145, 2022). Therefore, organic pollution from the ambient air hardly changes the wettability of the laser-ablated aluminum surface. In particular, all samples were hermetically stored to reduce pollution from the laboratory's ambient air before experimental characterizations.

The following image shows part of the samples in our experiments, for reference:

Fig. R2.1. Part of the samples in our experiments for different testing projects.

The supplemented contents in the methods of the main text are shown as:

Multiple samples were tested for each condition to ensure that the surfaces are not contaminated, damaged or otherwise altered in consecutive experiments. We also separately prepared different groups of specimens

for multiple testing projects such as measurements of wettability, high-speed imaging of droplet dynamic behaviors, scanning electron microscopy imaging, laser confocal scanning of morphology and microfocus-beam X-ray diffraction to avoid mutual interference between different testing projects.

Comment 6:

The microstructural characteristics is formed under the extreme condition where femtosecond laser shines on the aluminum-based surface with the ultrashort temporal scales (~10-15s) and the high energy density (>1014W/cm²). The flocculent structures are formed on the micro-protrusion due to the laser pulse ablation and the resolidification of ejected particles. The obvious differences between places near and far away the laser-scanned areas can be observed from the scanning electron microscope images of laser-ablated microstructures with large Ds (Supplementary Fig. 3 and 4). In addition, during the line-by-line laser scanning process, the aluminum substrate is also thermally oxidized, which will lead to the fact that oxidized aluminum films are formed. The authors should give the localized XRD results on the laser-scanned grooves and flat block.

Response:

We thank the reviewer for the detailed comment. We have supplemented the localized X-ray diffraction tests and added the localized X-ray diffraction results on the laser-scanned grooves and flat block. We used the 6061 aluminum sheet with a thickness of 0.3mm (thinner sheets may be pierced by the laser according to the height of the laser-ablated structure) in order to minimize the interference of the substrate layer on the X-ray diffraction of the laser-ablated layer. The average grain size of the used 6061 aluminum substrate was 18.25 μm as shown in Fig. R2.2 (Zhao S. T. et al. Mater. Des. 223, 111145, 2022), and the focal size of micro-XRD was about 50 μm . Thus, accurately distinguishing the effect of laser ablation on grain characteristics is difficult. Striped grooves were ablated in a single direction, and flat bars were distributed between the grooves as shown in Supplementary Fig. 8a for accurately identifying the X-ray diffraction results of grooves and flats. Meanwhile, continuous X-ray diffraction along the vertical direction of the laser-ablated grooves as shown in Supplementary Fig. 8a ensured the acquisition of crystallographic features of the groove and flat.

X-ray diffraction peaks of aluminum appeared at 38.88°, 45.38°, corresponding to the aluminum crystal orientation of 111 and 200, respectively. The intensity of diffraction peaks at 38.88°, 45.38° continuously changed during the continuous testing process, indicating the difference in the crystallographic characteristics between the laser-ablated groove and the flat as shown in Supplementary Fig. 8b. Specifically, the intensity of the diffraction peak at 38.88° continuously increased while the intensity of the diffraction peak at 45.38° continuously decreased, indicating that the thermal effect of the laser induced a transition in crystal orientation. We supplemented confirmatory testing on the edge of the same substrate to verify the intensity of X-ray diffraction peaks corresponding to laser-ablated grooves and the flat between grooves as shown in Supplementary Fig. 8a. The selected testing point was far from the laser ablation area and the same substrate ensured the same original crystal characteristics, avoiding the concentrated thermal effect of the laser and the differences in original crystal characteristics of different substrates, respectively. The temperature rise was

limited and could not significantly change the crystal characteristics at the edges of the substrate although the temperature of the substrate also integrally increased during the progress of laser ablation. Therefore, the X-ray diffraction results of grooves and flats could be distinguished through confirmatory testing at the edge of the substrate.

The diffraction peak corresponding to the aluminum crystal orientation of 111 was relatively weaker than the diffraction peak corresponding to the aluminum crystal orientation of 200 for the comparison point as shown in Supplementary Fig. 8c. Specially, the diffraction peak corresponding to the aluminum crystal orientation of 200 appeared at a slightly larger angle of 45.63° , indicating that the laser induced the left shift of the diffraction peak corresponding to the aluminum crystal orientation of 200. Hence, laser ablation induced a transition of the aluminum crystal orientation at grooves from 200 to 111 compared to the flat and the substrate. Similar conclusions of the aluminum crystal orientation can also be found from previous studies as shown in Fig. R2.3 (Zhao S. T. et al. Mater. Des. 223, 111145, 2022). The progress of laser ablation also induced lattice distortion at both grooves and flats in laser-treated area compared to the edges of the substrate.

Fig. R2.2. The distribution of the grain size of the selected 6061 aluminum substrate. (Zhao S. T. et al. Mater. Des. 223, 111145, 2022)

Fig. R2.3. XRD results of the original substrate, the laser-ablated substrate, and the heat-treated substrate after laser ablation. (Zhao S. T. et al. Mater. Des. 223, 111145, 2022)

The supplemented figure in Supplementary Information is shown as:

Supplementary Figure 8 | Localized X-ray diffraction results on the laser-scanned grooves and flats. a, Diagram of continuous X-ray diffraction scanning across one groove and flats. **b,** The variation of X-ray diffraction peaks with continuous movement of testing position. **c,** Selected X-ray diffraction results of the flat (point 1), the groove (point 7) and the comparison point.

The supplemented discussion in Supplementary Information is shown as:

S2. The thermal effect of laser ablation on aluminum grains

We used the 6061 aluminum sheet with a thickness of 0.3mm (thinner sheets may be pierced by the laser according to the height of the laser-ablated structure) in order to minimize the interference of the substrate layer on the X-ray diffraction of the laser-ablated layer. The average grain size of the used 6061 aluminum substrate was 18.25 μm , [5] and the focal size of micro-XRD was about 50 μm . Thus, accurately distinguishing the effect of laser ablation on grain characteristics is difficult. Striped grooves were ablated in a single direction,

and flat bars were distributed between the grooves as shown in Supplementary Fig. 8a for accurately identifying the X-ray diffraction results of grooves and flats. Meanwhile, continuous X-ray diffraction along the vertical direction of the laser-ablated grooves as shown in Supplementary Fig. 8a ensured the acquisition of crystallographic features of the groove and flat.

X-ray diffraction peaks of aluminum appeared at 38.88° , 45.38° , corresponding to the aluminum crystal orientation of 111 and 200, respectively. The intensity of diffraction peaks at 38.88° , 45.38° continuously changed during the continuous testing process, indicating the difference in the crystallographic characteristics between the laser-ablated groove and the flat as shown in Supplementary Fig. 8b. Specifically, the intensity of the diffraction peak at 38.88° continuously increased while the intensity of the diffraction peak at 45.38° continuously decreased, indicating that the thermal effect of the laser induced a transition in crystal orientation. We supplemented confirmatory testing on the edge of the same substrate to verify the intensity of X-ray diffraction peaks corresponding to laser-ablated grooves and the flat between grooves as shown in Supplementary Fig. 8a. The selected testing point was far from the laser ablation area and the same substrate ensured the same original crystal characteristics, avoiding the concentrated thermal effect of the laser and the differences in original crystal characteristics of different substrates, respectively. The temperature rise was limited and could not significantly change the crystal characteristics at the edges of the substrate although the temperature of the substrate also integrally increased during the progress of laser ablation. Therefore, the X-ray diffraction results of grooves and flats could be distinguished through confirmatory testing at the edge of the substrate.

The diffraction peak corresponding to the aluminum crystal orientation of 111 was relatively weaker than the diffraction peak corresponding to the aluminum crystal orientation of 200 for the comparison point as shown in Supplementary Fig. 8c. Specially, the diffraction peak corresponding to the aluminum crystal orientation of 200 appeared at a slightly larger angle of 45.63° , indicating that the laser induced the left shift of the diffraction peak corresponding to the aluminum crystal orientation of 200. Hence, laser ablation induced a transition of the aluminum crystal orientation at grooves from 200 to 111 compared to the flat and the substrate. Similar conclusions of the aluminum crystal orientation can also be found from previous studies. [5] The progress of laser ablation also induced lattice distortion at both grooves and flats in laser-treated area compared to the edges of the substrate.

The supplemented reference in Supplementary Information is shown as:

[5] Zhao, S. T. et al. Efficient fabrication of ternary coupling biomimetic superhydrophobic surfaces with superior performance of anti-wetting and self-cleaning by a method. *Mater. Des.* 223, 111145 (2022).

The supplemented contents in results of the main text are shown as:

The aluminum substrate was also thermally oxidized during the line-by-line laser scanning process, which could lead to the fact that oxidized aluminum films were formed. We discussed the localized X-ray diffraction results (Supplementary Fig. 8) on the laser-scanned grooves and flats in Supplementary Information S2.

The supplemented contents in method of the main text are shown as:

4.3. Localized X-ray diffraction on the laser-scanned grooves and flats

Wide angle X-ray diffraction (WAXD) experiments were conducted on a customized microfocus X-ray diffraction system (Xenocs SA, France) in transmission mode with Cu K α radiation. The X-ray radiation wavelength was 0.154 nm, and the beam diameter of the X-ray radiation at the focal position was about 50 μ m. Each WAXD pattern was exposed for 120 s at a sample-to-detector distance of 29 mm using a Pilatus 100 K detector (Dectris, Switzerland).

Comment 7:

The objective of section profiles in Supplementary Fig. 6c is to show that the structure height do not significantly change under different laser scan spacings. However, it is very difficult to get the data of structure height from the figure by using normal linear ordinate. The authors should give a table to directly exhibit the geometry parameters.

Response:

We thank the reviewer for the helpful suggestion. We have calculated average structure heights of all surfaces based on the data in Fig. S6c, and added the table to directly exhibit the structure heights. The laser scan speed which could change the structure height was constant in our experiment, and the change in laser scan spacing could not significantly affect the structure height. However, structure height was slightly different in our experiments due to the laser processing technology. The structure height can qualitatively affect the droplet rebound behavior, and thus affect the contact time of the droplet. Specific structure height, especially large structure heights, can induce pancake bouncing with significant short contact time of the droplet, and the droplet cannot rebound on structures with too small height (Liu Y. H. et al. Nat. Phys. 10, 515-519, 2014) (Pan W. H. et al. Nanoscale 13, 14023-14034, 2021) (Song Y. X. et al. Opt. Laser Technol. 102, 25-31, 2018) (Song J. L. et al. ACS Nano 11, 9259-9267, 2017). But our structure height almost all less than 100 μ m, significantly lower than the structure height that induces pancake bouncing of droplet. Meanwhile, pancake bouncing did not occur on our laser-ablated structures, and the structures are high enough to support the rebound of the droplet. Therefore, the slight differences in structure heights of different surfaces have almost no significant effect on the rebound dynamics of droplets.

The optimized Supplementary Fig. 7 is shown as:

Supplementary Figure 7 | Geometric scanning of laser-ablated microstructures with a series of D_s . a-b, Three-dimensional image and section profile of S500 surface. The D_s was basically consistent with the set laser scanning spacing, and the average height difference between peak and valley of laser ablated grooves was 62.4 μm . **c,** Section profiles of microstructures with D_s of 50-1000 μm . Different laser scan spacings did not significantly change the structure height, and the slight differences in structure heights of different surfaces have almost no effect on the rebound dynamics of droplets.

Comment 8:

The authors stated the “droplet trampoline” effect on metal-based superhydrophobic surface is investigated in this paper. However, there is no figure to show the superhydrophobicity of metal-based surfaces. In addition, is there a mode transition from Cassie-Baxter to Wenzel state with the continuous increase of D_s as the sudden increase of sliding angle occurs at the D_s of 500 μm .

Response:

We thank the reviewer for the important comment. Significant supporting images of contact angle, slide angle, advancing contact angle and receding contact angle have been added to display wettability of surfaces

with different spacings more intuitively. The contact angles of S50-S1000 surfaces were all $>150^\circ$, confirming the superhydrophobicity of the metal-based surfaces. The contact angle of the surface decreased and the adhesion enhanced with the continuous increase of the structure spacing due to the reduced ablation area fraction.

The penetration induced by wetting pressure can cause the transition of the contact state between droplets and local particles from the Cassie-Baxter state to the Wenzel state (Lafuma, A., Quere, D. *Nat. Mater.* 2, 457-460, 2003). The transition from Cassie-Baxter state to Wenzel state occurred on surfaces with relatively small structure spacing due to the permeation effect induced by large vertical dynamic pressure of the droplet. As the structure spacing increased, droplets were easier to spread laterally, and the fluctuation of the surface decreased, weakening the local penetration effect. However, the adhesion effect occurred on the surface with large structure spacing due to large flats that were not ablated. For the liquid residue induced by adhesion effect on the surface with large structure spacing, the contact angle of the adhered droplet was close to the intrinsic contact angle of the surface, which was different from the permeation effect on the surface with small structure spacing. The transition from Cassie-Baxter state to Wenzel state was difficult to occur on the surface with relatively large structure spacing due to the release of the dynamic pressure from the impacting droplet. Droplets could not slide from the surface with the structure spacing $\geq 500 \mu\text{m}$ due to enhanced adhesion caused by sparse microstructures.

The optimized Supplementary Fig. 10 is shown as:

Supplementary Figure 10 | Wettability characteristics of laser-ablated microstructures with different D_s .

a, The reduced contact angle (CA) and the increased sliding angle (SA), indicating the weakened water repellency with the continuous increase of D_s . **b**, The advancing contact angle (ACA), receding contact angle (RCA) and calculated contact angle hysteresis (CAH). Large D_s aggravated the heterogeneity of wettability and increased the movement resistance of solid-liquid-air contact lines. **c-f**, Significant supporting images and results for the wettability characterization.

Comment 9:

Do the authors take into account the effect of energy dissipation caused by the partial penetration on the number of consecutive droplet rebounds. As the authors stated, the occurrence of partial penetration obviously inhibited the consecutive rebound of droplet.

Response:

Thank the reviewer for the comment. We have taken into account the effect of energy dissipation caused by the partial penetration on the number of consecutive droplet rebounds (N_R). On the one hand, dense microstructures could facilitate the occurrence of partial penetration. On the other hand, the large impact force of the droplet at high We could aggravate the partial penetration and increase the viscous dissipation of the droplet. The partial penetration of the droplet to microstructures was unpredictable during each droplet impact process, even if the structure spacing and We were the same. Therefore, it is difficult to quantitatively consider the energy dissipation caused by partial penetration in a single process of consecutive droplet rebounds. However, the effect of energy dissipation caused by the partial penetration on the N_R can be qualitatively analyzed. Small structure spacing could increase the pin dissipation, and high We could enhance the viscous dissipation, thereby inhibiting the conservative rebound of droplet according to the proposed formula of N_R .

Comment 10:

The parameter of maximum wetted diameter (D_{wetted}) rather than the maximum spreading diameter (D_{max}) represent the diameter of pinned area, as the pin dissipation energy is originated from the crawling movement of the contact line during the spreading and contracting process.

Response:

We agree with the reviewer's viewpoint. The parameter of maximum wetted diameter (D_{wetted}) rather than the maximum spreading diameter (D_{max}) represent the diameter of contact area between the droplet and the surface. We have supplemented the D_{wetted} of droplets on the S50, S500, and S1000 surfaces as typical surfaces, as shown in Supplementary Fig. 11. The D_{wetted} of droplets approximately conformed to $0.82We^{0.25}$.

The Weber number We represents the ratio of the effects of inertial force and surface tension. The inertial force was much smaller than the surface tension at extremely low We , and the surface tension significantly controlled the elliptical shape of droplets at maximum spread. Conversely, the inertial force gradually enhanced and played a dominant role as We increased, and the shape of the droplets tends towards pancakes. Therefore, the difference between D_{wetted} and D_{max} was relatively obvious at extremely low We , while the difference was not significant and could be ignored at high We . Even so, The D_{wetted} of droplets approximately conformed to $0.85We^{0.25}$.

The maximum spreading diameter D_{max} has been replaced with the maximum wetted diameter D_{wetted} in method of the main text.

The revised formulas are shown as:

$$N_R = N\varepsilon = N(1 - 0.284\frac{A_D}{D_S}We^{-0.5} - 0.021We^{0.75} - 1.14 \times 10^6 C_D D_F^2 We^{-1}) \quad (1)$$

$$N_R = N(1 - \frac{A}{D_S}We^{-0.5} - BWe^{0.75} - C(D_S - 500)^2 We^{-1}) \quad (2)$$

$$E_P \approx \frac{A_D}{D_S} \left[\int_0^{R_{\max}} 2\pi\gamma_{LA} (\cos\theta - \cos\theta_A) r dr + \int_0^{R_{\max}} 2\pi\gamma_{LA} (\cos\theta_R - \cos\theta) r dr \right]$$

$$= \frac{A_D}{D_S} \pi\gamma_{LA} (\cos\theta_R - \cos\theta_A) R_{\text{wetted}}^2 \quad (3)$$

$$E_P \approx 2.27 \times 10^{-8} \frac{A_D}{D_S} We^{0.5} \quad (4)$$

The revised Supplementary Fig. 11 is shown as:

Supplementary Figure 11 | Maximum wetted factors (β_{wetted}) of selected microstructures with three typical D_S as a function of We . Experimental factors β_{wetted} of three surfaces were close to $\beta_{\text{wetted}}=0.85We^{0.25}$. Droplets were easier to spread on the surfaces of S500 and S1000 and had greater wetting diameters especially for high We due to the weakening of pinning resistance.

Comment 11:

As the authors stated, the parameter DF in equation 6 (Supplementary Information S2) is the width of the adhesive area on the hydrophilic flat. How does the authors confirm that the critical size for the hydrophilic flat is 500 μm . In addition, the authors stated that the final contact area of residual liquid is hydrophilic region, which will raise a question whether the un-scanned areas remains intrinsic hydrophilicity.

Response:

We thank the reviewer for the valuable comment. The structure spacing of 500 μm was the critical structure spacing at which droplets could not statically slide on the surface, indicating that the hydrophilic flat was large enough to adsorb droplets and hinder their separation from the surface. The un-scanned area (flat) remained intrinsic hydrophilicity because there was almost no microstructure on the un-scanned flat. Meanwhile, the contact angle of the residual droplet on flat was close to the intrinsic contact angle of the substrate, indicating that the un-scanned area was intrinsically hydrophilic.

The revised contents in discussion of the main text are shown as:

The D_S of 500 μm was the critical structure spacing at which droplets could not statically slide on the surface, indicating that the hydrophilic flat was large enough to adsorb droplets and hinder their separation from the surface. Meanwhile, the adhesive area usually appeared only when the width of the hydrophilic flat increased to over the critical size (500 μm) according to the experimental results. Thus, D_F can be expressed as $D_F = D_S - 500$, and the N_R can be further described as:

$$N_R = N \left(1 - \frac{A}{D_S} We^{-0.5} - B We^{0.75} - C (D_S - 500)^2 We^{-1} \right) \quad (2)$$

The optimized Supplementary Fig. 10 is shown as:

Supplementary Figure 10 | Wettability characteristics of laser-ablated microstructures with different D_S .

a, The reduced contact angle (CA) and the increased sliding angle (SA), indicating the weakened water

repellency with the continuous increase of D_s . **b**, The advancing contact angle (ACA), receding contact angle (RCA) and calculated contact angle hysteresis (CAH). Large D_s aggravated the heterogeneity of wettability and increased the movement resistance of solid-liquid-air contact lines. **c-f**, Significant supporting images and results for the wettability characterization.

Comment 12:

When We exceeds 43, the number of consecutive droplet rebounds is zero on S200 surface (Supplementary Fig. 11), indicating that no successive droplet rebound occurs. The increase of We number obviously increases the dynamic pressure, which causes the fact that droplet can easily penetrate into the microstructures. The penetration effect produced by the increased We number is misleading qualitatively, which weakens their conclusion on S200 surface.

Response:

We thank the reviewer for the valuable comment. Flat appeared due to the inability of the ablation area to fully cover the surface when the laser scan spacing was larger than 50 μm . Hydrophilic flats on S100 and S200 surfaces facilitated penetration effect and droplet residue, thereby inhibiting the consecutive droplet rebound. Meanwhile, for the S100 and S200 surfaces, the structure spacings were still very small, inducing pin effect. Therefore, the S100 and S200 surfaces have two disadvantages: small structure spacing and hydrophilic flat, which are unfavorable for consecutive droplet rebound.

The large dynamic pressure brought by relatively high We can facilitate penetration effect, which has been confirmed (Lafuma, A., Quere, D. *Nat. Mater.* 2, 457-460, 2003). As shown in Fig. R2.4, the advancing contact angle, which is related to surface hydrophobicity, obviously declines as the contact pressure increases. Especially with the increase of wetting pressure, the receding contact angle significantly declines, exhibiting the serious permeation effect and adhesion. Therefore, S100 and S200 surfaces exhibited worse consecutive rebound performance of droplets compared to S50 surface at relatively high We . However, the numbers of consecutive droplet rebound of S100 and S200 surfaces were still higher than those of S50 surface at relatively low We , confirming the weakening of pin effect by increasing structure spacing, as shown in Fig. S7.

Fig. R2.4. Wetting state transition from Cassie-Baxter state to Wenzel state induced by contact pressure,

exhibiting large droplet dynamic pressure to promote permeation effect and disrupt water repellency. Left image: The apparent contact angle θ^* is measured as a function of the imposed pressure ΔP (open points). The upper and lower dotted lines are, respectively, the values measured for a large drop initially formed either by deposition or condensation. Right image: The receding angle θ_r^* , observed after imposing and relaxing a pressure ΔP , is plotted as a function of ΔP (open points). The filled points correspond to drops deposited on a single surface, and ΔP is then the pressure applied by the drop on the surface; its variation is obtained by taking different drop sizes. The upper and lower dotted lines respectively indicate the value of θ_r^* for a drop deposited on the substrate or obtained by condensing a vapour. (Lafuma, A., Quere, D. Nat. Mater. 2, 457-460, 2003)

The revised contents in Supplementary Information are shown as:

Specifically, abnormal phenomenon of N_R at relatively high We occurred on S100 and S200 surfaces (especially on the S200 surface). Flat appeared due to the inability of the ablation area to fully cover the surface when the laser scan spacing was larger than 50 μm . Hydrophilic flats on S100 and S200 surfaces facilitated penetration effect and droplet residue, thereby inhibiting the consecutive droplet rebound. Meanwhile, for the S100 and S200 surfaces, the structure spacings were still very small, inducing pin effect. As a result, the S100 and S200 surfaces have two disadvantages: small structure spacing and hydrophilic flat, which are unfavorable for consecutive droplet rebound. The large dynamic pressure brought by relatively high We can facilitate penetration effect, which has been confirmed. [6] Therefore, S100 and S200 surfaces exhibited worse consecutive rebound performance of droplets compared to S50 surface at relatively high We . However, the numbers of consecutive droplet rebound of S100 and S200 surfaces were still higher than those of S50 surface at relatively low We , confirming the weakening of pin effect by increasing structure spacing, as shown in Supplementary Fig. 9.

The supplemented reference in Supplementary Information is shown as:

[6] Lafuma, A., Quere, D. Superhydrophobic states. Nat. Mater. 2, 457-460 (2003).

Again, we are so sorry to bring you so much trouble because of our carelessness. All the detailed changes are marked in the revised manuscript. At last, thank you for your review and your comments again.

Best regards.

Luquan Ren, Professor, Academician of Chinese Academy of Sciences, Jilin University

Hongwei Zhao, Professor, Vice President, Jilin University

Zhichao Ma, Professor, Deputy Dean, School of Mechanical and Aerospace Engineering, Jilin University

REVIEWERS' COMMENTS

Reviewer #1 (Remarks to the Author):

The authors have resolved all my questions.

Reviewer #2 (Remarks to the Author):

The revised paper had addressed my comments, which lead me suggest that this paper is suitable for publication in Nature Communications.